# NePuDA: Neighborhood-Purifying Discriminant Analysis

## Abstract

Linear Discriminant Analysis (LDA) is a popular technique for supervised dimensionality reduction due to its clarity and interpretability. However, LDA and its variants often assume that data within each class are Gaussian-distributed or form distinct groups/subclasses, which is not always true for high-dimensional, real-world datasets, where classes can have complex and irregular shapes and exhibit significant overlap. Recognizing this limitation, we propose a novel approach, **Ne**ighborhood-**Pu**rifying **D**iscriminant **A**nalysis, which forgoes the search for an ideal, class-separated subspace in favor of one where data samples are naturally surrounded by neighbors from the same class. Specifically, NePuDA aims to identify projection directions that reinforce this neighborhood purity for all data samples, with the intuitive logic that if an object shares characteristics with a known category, it likely belongs to that category. Accordingly, we formulate the objective function of the proposed method and introduce an iterative optimization procedure to solve it in an efficient manner. Detailed theoretical analyses are provided, covering convergence, computational complexity, and connections to existing LDA variants. Extensive empirical evaluations on a range of synthetic and real-world datasets demonstrate that NePuDA consistently extracts highly discriminative features, outperforming twelve classical and state-of-the-art supervised dimensionality reduction algorithms in classification accuracy. Our code is publicly available at https://anonymous.4open.science/r/NePuDA_code-C47F/.

## 1 Introduction

With the rapid growth of information technology, high-dimensional data have become increasingly prevalent in practical applications. While these data often carry rich information, their processing and analysis are quite challenging due to the so-called curse of dimensionality (Altman & Krzywinski, 2018). Dimensionality reduction (DR), as an effective technique to alleviate this problem, aims to learn the low-dimensional representation of high-dimensional observations based on given criteria (Oikarinen et al., 2019), so as to enhance the accuracy and efficiency of subsequent data analytics tasks (Wang et al., 2022a; 2024b; Verhaeghe et al., 2022). Over the years, DR techniques have been widely adopted in a variety of fields, including but not limited to, face recognition (Zhao et al., 2019), text retrieval (Kim et al., 2005), and video processing (Su et al., 2017). More recently, deep models, including deep neural networks and large language models, have demonstrated their strength in representation learning. Integrating DR with these models can further boost their performance for several key reasons. First, high-dimensional data are generally believed to lie near a lower-dimensional manifold (Bengio et al., 2013). DR helps uncover and exploit this structure, thereby reducing complexity and enhancing both the effectiveness and computational efficiency of subsequent deep models (Wang et al., 2024a). Consistently, recent work shows that DR can improve the compactness and efficiency of language-model representations, from reducing the dimensionality of pretrained sentence embeddings while largely preserving downstream performance (Zhang et al., 2024). Besides, it can also compress high-dimensional LLM-generated representations for effective and efficient downstream recommendation (Ma et al., 2026), and reduce LLM activation dimensionality for model compression and faster inference (Sakr & Khailany, 2024). Second, DR can strip away redundant information and suppress noise in the training data (Saberi-Movahed et al., 2025). Since deep models are often sensitive to noisy or redundant inputs (Han et al., 2018; Li et al., 2021), while LLMs are likewise highly affected by data quality (Zhang et al., 2025), this cleaning step aligns well with their training needs. Third, by revealing the intrinsic geometry or structure

hidden in the data, DR provides a promising avenue for improving the interpretability and explainability of deep models (Cunningham & Ghahramani, 2015). In recent years, DR has been increasingly embedded within deep architectures, leading to impactful applications in areas such as feature extraction (Saberi-Movahed et al., 2025), model compression (Sakr & Khailany, 2024), and online learning (Alvarado-Perez et al., 2025).

DR techniques can be broadly classified into two categories based on the availability of label information: unsupervised and supervised methods. Unsupervised methods operate without reference to external labels, focusing solely on uncovering and preserving the intrinsic structure of data. Notable examples of unsupervised DR techniques include Principal Component Analysis (PCA) (Hotelling, 1933), Multi-Dimensional Scaling (MDS) (Torgerson, 1952), and Isometric feature Mapping (IsoMap) (Tenenbaum et al., 2000). In contrast, supervised DR methods utilize label information to guide the DR process, aiming to maintain the class distinctions within the low-dimensional representation. This attribute makes supervised DR methods particularly advantageous for discriminative analysis tasks where maintaining the separation between different classes is crucial.

Linear Discriminant Analysis (LDA) (Fisher, 1936) is the most pioneering method developed for supervised DR. It looks for the projection direction that maximizes the between-class distance while concurrently minimizing the within-class distance. Although LDA has found widespread application across diverse fields, it operates on certain assumptions that limit its applicability. Specifically, LDA presupposes that the data within each class are normally distributed, and it employs the mean of each class to represent the entire class in the process of maximizing the distance between classes. Furthermore, LDA treats each class as a single entity, focusing on minimizing the within-class variance in the reduced subspace. This setting is effective for well-separated, normally distributed classes but may not be suitable for data with more complex structures, which frequently occur in real-world scenarios.

To overcome the limitations of traditional LDA, a variety of enhanced LDA models have been introduced in recent years. The ideas of these advanced versions of LDA can be broadly classified into three main categories: metric modification, max-min strategy, and neighborhood exploration. Metric modification involves redefining the computation of between/within-class scatters (e.g., subclass discriminant analysis (SDA) (Zhu & Martinez, 2006) and geometric mean-based scatter optimization (Tao et al., 2008)), reformulating the trace ratio objectives (e.g., ratio sum LDA (Wang et al., 2022a) and quadratic form of trace ratio LDA (Wang et al., 2022b)), and altering the distance measurements (e.g., Wasserstein discriminant analysis (Flamary et al., 2018; Liu et al., 2020), $\ell_1$-norm based LDA (Zhong & Zhang, 2013), $\ell_{2,1}$-norm based LDA (Zhao et al., 2019; Nie et al., 2021)). The second category, max-min strategy, aims to enhance class discriminability by maximizing the distance between the closest opposing classes while simultaneously minimizing the distance between the most separated points within the same class. Notable methodologies employing this strategy include worst-case LDA (Zhang & Yeung, 2010), heteroscedastic max–min distance analysis (Su et al., 2018), and worst-case discriminative dimensionality reduction (Wang et al., 2024b). By prioritizing the differentiation of nearby or even overlapping classes and tightening the cohesion within individual classes, the max-min strategy has demonstrated its effectiveness in resolving classification challenges presented by complex datasets. The final category, neighborhood exploration, places emphasis on the local distribution of data by modeling the relationships between neighboring data points, which enables the characterization of the geometric structure of the dataset and the boundary of classes. Representative methods in this category include neighborhood component analysis (NCA) (Goldberger et al., 2004), local Fisher discriminant analysis (LFDA) (Sugiyama, 2007), large margin nearest neighbor (LMNN) (Weinberger & Saul, 2009), local LDA (LLDA) (Kim & Kittler, 2005), and neighborhood minmax projections (NMMP) (Nie et al., 2007; Zhao et al., 2018). Please refer to Appendix A for more technical details of LDA and its variants.

The variants of LDA discussed above have achieved remarkable success in supervised DR tasks. Yet, they exhibit limitations in capturing the subtle characteristics of datasets for discrimination purposes. Both the metric modification and max-min strategies, while relaxing the assumption in classic LDA that each class is normally distributed, continue to operate at the class or subclass level. Similar to traditional LDA, these methods use the mean of each class or subclass to typify the data group. This coarse-scale perspective can overlook critical fine-grained information necessary for discriminative analysis. In contrast, neighborhood exploration methods pay explicit attention to local data structures. However, they use an

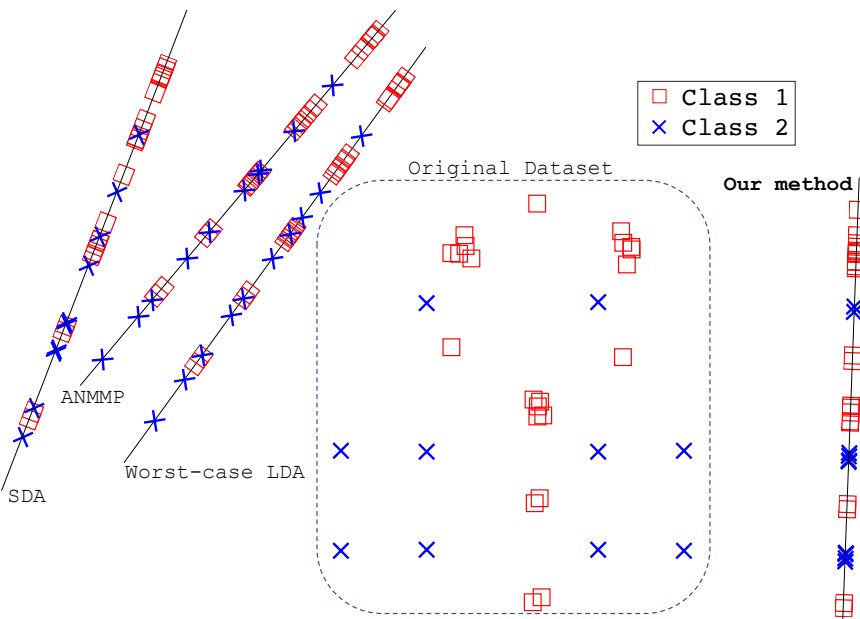

Figure 1: A schematic illustration of the proposed method. The 2-dimensional dataset consists of two intertwining classes, denoted by red squares and blue crosses, respectively. The 1-dimensional projection results of the proposed method and three representative variants of LDA, i.e., SDA (Zhu & Martinez, 2006), worst-case LDA (Zhang & Yeung, 2010), and NMMP (Zhao et al., 2018), are provided.

average distance/scatter as their learning objective, either across all points in a local neighborhood or over all pairs of nearby points. When averaging is used as the primary criterion for learning, these methods inherently prefer projections that preserve the status quo for neighboring data points that are already well-separated between classes and closely packed within classes in the original feature space. Such a preference can disadvantage 'at-risk' points, those that have neighbors from different classes, by inadequately addressing their positioning. Consequently, this can lead to suboptimal projection results, thus negatively impacting the effectiveness of these methods for subsequent discriminative analysis tasks.

To address the limitations of current methodologies, in this paper, we present a novel supervised DR method dubbed **Ne**ighborhood-**Pu**rifying **D**iscriminant **A**nalysis (NePuDA). The proposed method departs from the conventional goal of finding a subspace where classes or subclasses are distinctly separated—a standard that is often unrealistically high. Instead, our goal is to uncover a subspace where each data point is naturally encircled by neighbors from its own class. This target is not only more feasible but also provides an adequate foundation for effective discriminative analysis. Furthermore, NePuDA distinguishes itself from existing neighborhood exploration methods that uniformly consider all neighborhoods by averaging across all data points or pairs within the objective function. Our innovative strategy focuses on identifying and prioritizing data points that are difficult to separate or are at risk of misclassification. Specifically, it aims to purify the neighborhood for each data point, creating a subspace where even those with mixed-class neighbors in the original feature space can enjoy a homogeneous class environment in their local neighborhood.

Figure 1 schematically demonstrates the concept behind our proposed method. The original 2-dimensional dataset consists of two classes that are intricately intertwined: one class is indicated by red squares, and the other by blue crosses. Each class contains 'at-risk' data points that are either in close proximity to or surrounded by samples from the opposing class, presenting difficulties for discriminant analysis. We compare the projection results of the proposed method with those of three LDA variants: SDA (Zhu & Martinez, 2006) from the metric modification category, worst-case LDA (Zhang & Yeung, 2010) from the max-min strategy category, and NMMP (Zhao et al., 2018) from the neighborhood exploration category. The left side of the figure shows that the projections generated by these LDA variants result in obvious overlaps between the classes, which pose challenges to the subsequent task of classification. These overlaps occur because the

existing LDA variants do not fully account for the local characteristics of individual data points, leading to the creation of a sub-optimal discriminative subspace. The proposed method, NePuDA, diverges from these approaches by purifying the neighborhood around each individual data point in the projected subspace, with particular attention to those 'at-risk'. This strategy allows NePuDA to achieve clearer separation between data points from different classes, as shown on the right-hand side of the figure. Such distinct separation improves the performance of classification tasks, even when a straightforward nearest neighbor classifier is employed.

## 1.1 Our Contributions

The main contributions of this paper can be summarized as follows.

1. We introduce a new idea, aiming at identifying and purifying the neighborhood of the most difficult-to-purify data points, thereby improving the lower bound of neighborhood purity in the learned subspace, for enhancing the supervised DR. By doing so, we aim to ensure that each individual data point, rather than the average over all data points within classes, is locally enveloped by neighbors belonging to the same class in the resulting subspace.

2. Building on this idea, we devise a novel supervised DR method called **Ne**ighborhood-**Pu**rifying **D**iscriminant **A**nalysis (NePuDA). We formulate the objective function of NePuDA, develop an iterative approach to solve the optimization problem, and establish the convergence properties of our iterative algorithm. We also assess the computational complexity of NePuDA, and connect it to other well-established supervised DR techniques.

3. We systematically validate the effectiveness of the proposed method by comparing its performance with 14 classical and state-of-the-art DR methods on both illustrative synthetic examples and 13 real-world datasets. Additionally, we further investigate the parameter sensitivity of the proposed method and confirm the rapid convergence of the proposed iterative process.

The structure of this paper is outlined as follows: In Section 2, the conceptual framework and objective function of the proposed method are detailed. Section 3 introduces the optimization strategy for the proposed method. Section 4 presents the theoretical analysis of the proposed method. Experimental validations on both synthetic and real-world datasets are reported in Section 5. Finally, we conclude our paper in Section 6.

## 2 Neighborhood-Purifying Discriminant Analysis

In this section, we introduce the concept of the proposed Neighborhood-Purifying Discriminant Analysis (NePuDA). First, we introduce some basic notations and definitions of our work. After that, we explain the construction of a pure neighborhood and the rationale behind our design. Following this, we present the formulation of the overall objective function for the proposed method.

### 2.1 Notations and Definitions

In this paper, we adopt a standard notation where matrices are represented by bold uppercase letters (e.g., '$\mathbf{A}$'), and vectors are indicated by bold lowercase letters (e.g., '$\mathbf{a}$'). We denote the trace of a matrix $\mathbf{A}$ as $\text{tr}(\mathbf{A})$. If $\mathbf{A}$ is positive semidefinite, we denote it as $\mathbf{A} \succeq \mathbf{0}$. For a vector $\mathbf{w}_i \in \mathbb{R}^d$, its $\ell_2$-norm is expressed as $\|\mathbf{w}_i\|_2 = \sqrt{\sum_{j=1}^{d} w_{ij}^2}$, where $w_{ij}$ denotes the $j$-th component of $\mathbf{w}_i$, and $d$ is the dimension of the feature space. For a set $\Omega$, its size or cardinality is represented by $|\Omega|$. For readers' convenience and easy reference, these notations, along with other frequently used symbols, are concisely summarized in Appendix A.1.

### 2.2 Pure Neighborhood Construction

To enhance the separability of the dataset in the learned subspace, the proposed NePuDA method aims to purify each data point's neighborhood, ensuring that all data points are surrounded by neighbors from the

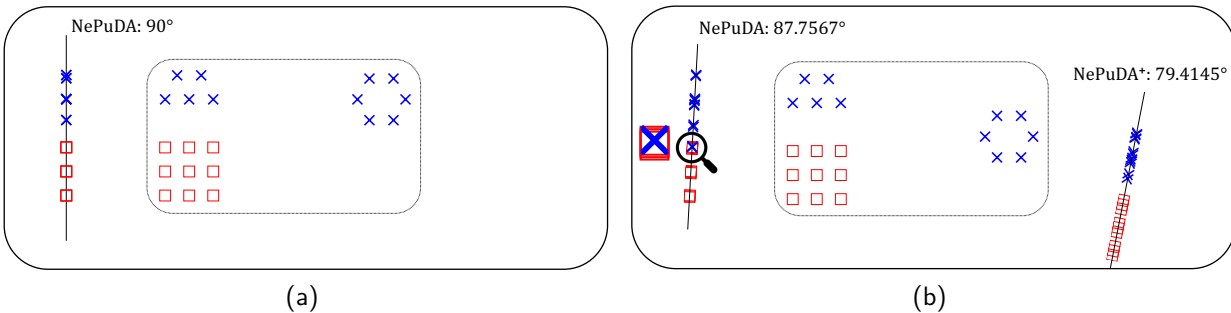

Figure 2: Illustration of the projection results of (a) NePuDA (pure neighborhood construction based on Eq. (1)) and (b) NePuDA$^+$ (pure neighborhood construction based on Eq. (2)).

same class. By doing so, NePuDA creates a more distinct subspace, which facilitates better classification performance.

Accordingly, we introduce the concept of pure neighborhood for each data point. Let $\mathbf{X} = \begin{bmatrix} \mathbf{x}_1, \mathbf{x}_2, \cdots, \mathbf{x}_n \end{bmatrix} \in \mathbb{R}^{D \times n}$ represent the data sample matrix, where each column vector $\mathbf{x}_i$ $(i = 1, \cdots, n)$ corresponds to a $D$-dimensional data sample, and $n$ denotes the sample number. Specifically, the pure neighborhood of $\mathbf{x}_i$ is defined as a set of data points that meet two criteria: 1) they are neighbors of $\mathbf{x}_i$, geometrically; and 2) they share the same label as $\mathbf{x}_i$. Mathematically, the pure neighborhood set of $\mathbf{x}_i$, denoted as $\mathcal{N}_i$, can be defined as follows:

$$\mathcal{N}_i = \{\mathbf{x}_j \mid \mathbf{x}_j \in \mathbb{N}_k(\{\mathbf{x}_\iota\}_{\iota=1}^n, \mathbf{x}_i) \wedge \ell_j = \ell_i\}, \tag{1}$$

where $\ell_i$ denotes the label of $\mathbf{x}_i$, and $\mathbb{N}_k(\{\mathbf{x}_\iota\}_{\iota=1}^n, \mathbf{x}_i)$ denotes the neighborhood set of $\mathbf{x}_i$ within the set $\{\mathbf{x}_\iota\}_{\iota=1}^n$ (excluding $\mathbf{x}_i$ itself), with $k$ being a positive integer representing the size of $\mathbb{N}_k$. The physical interpretation of this definition is that it selectively identifies homogeneous neighbors from the set of all geometric neighbors, effectively forming within-class neighborhoods. The objective of establishing such purified neighborhoods in the subspace is to mitigate the problem of class entanglement, which often arises due to the presence of mixed-class neighbors of data points in the original feature space. The set $\mathcal{N}_i$ defined in Eq. (1) demonstrates its effectiveness in certain scenarios, as illustrated in Fig. 2a. However, its performance may be suboptimal in other contexts, such as the case depicted in Fig. 2b. Specifically, in the dataset presented in Fig. 2b, while the construction approach described in Eq. (1) successfully aggregates same-class neighboring samples for a given data point in the original feature space, it may not necessarily generate a neighborhood-purified subspace. This limitation arises because data points from different classes, initially distant in the original space, may become neighbors or even entangled after the projection. To enhance the neighborhood purity for the learned subspace, we extend the definition of pure neighborhood set as follows:

$$\mathcal{N}_i^+ = \{\mathbf{x}_j \mid \mathbf{x}_j \in \mathbb{N}_k(\{\mathbf{x}_\iota\}_{\iota=1,\ldots,n}^{\ell_\iota = \ell_i}, \mathbf{x}_i)\}. \tag{2}$$

In contrast to the definition of $\mathcal{N}_i$ in Eq. (1), which selects same-class neighbors from the $k$ nearest neighbors (irrespective of classes), the above definition of $\mathcal{N}_i^+$ in Eq. (2) mandates the inclusion of $k$ nearest neighbors exclusively from the same class. This expands the neighborhood set by incorporating same-class data points that might be relatively distant in the original space. By enforcing a stronger intra-class cohesion, $\mathcal{N}_i^+$ potentially mitigates the risk of inter-class entanglement after the projection, thus enhancing the neighborhood purity in the learned subspace[1].

In addition to the construction of the pure neighborhood set, which is integral to the calculation of the within-class scatter calculation, we define the between-class neighborhood, which is necessary for the formulation of the between-class scatter. Specifically, the between-class neighborhood set, denoted as $\mathcal{N}_i^c$, is defined as

---

[1]Note that in our paper, NePuDA represents the proposed method that uses Eq. (1) to construct the pure neighborhood set $\mathcal{N}_i$, and NePuDA$^+$ denotes the enhanced version of the method, which utilizes Eq. (2) to construct the pure neighborhood set $\mathcal{N}_i^+$.

the complement of $\mathcal{N}_i$ with respect to $\mathbb{N}_k(\{\mathbf{x}_\iota\}_{\iota=1}^n, \mathbf{x}_i)$:

$$\mathcal{N}_i^c = \mathbb{N}_k(\{\mathbf{x}_\iota\}_{\iota=1}^n, \mathbf{x}_i) \setminus \mathcal{N}_i = \{\mathbf{x}_j \mid \mathbf{x}_j \in \mathbb{N}_k(\{\mathbf{x}_\iota\}_{\iota=1}^n, \mathbf{x}_i) \wedge \ell_j \neq \ell_i\}. \tag{3}$$

This formulation ensures that the between-class neighborhood set $\mathcal{N}_i^c$ comprises those $k$-nearest neighbors of $\mathbf{x}_i$ that do not belong to its pure neighborhood set $\mathcal{N}_i$, thus capturing the inter-class relationship in the neighborhood for discriminant analysis. Note that for both the standard pure neighborhood set $\mathcal{N}_i$ and its enhanced version $\mathcal{N}_i^+$, we employ a common between-class neighborhood set $\mathcal{N}_i^c$, as defined in Eq. (3). The purpose here is to identify data points that are close to the point $\mathbf{x}_i$ in the original feature space but belong to different classes, and to increase the separation between these nearby, different-class data points and the reference point $\mathbf{x}_i$ after projection into the low-dimensional subspace.

## 2.3 Objective Function of NePuDA

In this subsection, we formulate the objective function of the proposed NePuDA method according to the pure neighborhood set ($\mathcal{N}_i$ or $\mathcal{N}_i^+$) and the between-class neighborhood set $\mathcal{N}_i^c$ defined in the last subsection. Specifically, the *within-class-neighborhood scatter* of $\mathbf{x}_i$ is defined as follows[2]:

$$\mathbf{S}_i = \sum_{\mathbf{x}_j \in \mathcal{N}_i \text{ or } \mathcal{N}_i^+} (\mathbf{x}_i - \mathbf{x}_j)(\mathbf{x}_i - \mathbf{x}_j)^T. \tag{4}$$

To purify each data point's neighborhood in the $d$-dimensional subspace ($d < D$), with particular emphasis on data points that are challenging to separate, our optimization objective is different from the conventional approach of summing the within-class neighborhood scatter matrices of all samples. Instead, we employ a max-min strategy to maximize the neighborhood purity for the hard-to-separate ones:

$$\min_{\mathbf{W}} \max_i \ \text{tr}(\mathbf{W}^T \mathbf{S}_i \mathbf{W}), \tag{5}$$

which prioritizes the most challenging cases. By such design, we ensure that even in the worst-case scenario, the separation in all neighborhoods remains acceptable.

To quantify the dispersion of data points belonging to different classes within a specified neighborhood, we further introduce the concept of the *between-class-neighborhood scatter*, which is derived from the between-class neighborhood set $\mathcal{N}_i^c$ as defined in Eq. (3):

$$\mathbf{\Sigma}_i = \sum_{\mathbf{x}_j \in \mathcal{N}_i^c} (\mathbf{x}_i - \mathbf{x}_j)(\mathbf{x}_i - \mathbf{x}_j)^T. \tag{6}$$

By aggregating information from all classes, a global between-class-neighborhood scatter is defined as follows:

$$\mathbf{\Sigma} = \sum_{i=1}^n \frac{|\mathcal{N}_i^c|}{n} \mathbf{\Sigma}_i. \tag{7}$$

Subsequently, we formulate a unified objective function for NePuDA that incorporates both within-class and between-class neighborhood information for all data points:

$$\max_{\mathbf{W}} \frac{\text{tr}(\mathbf{W}^T \mathbf{\Sigma} \mathbf{W})}{\max_i \text{tr}(\mathbf{W}^T \mathbf{S}_i \mathbf{W})}, \quad \text{s.t. } \mathbf{W}^T \mathbf{W} = \mathbf{I}_d, \tag{8}$$

where the orthonormality constraint introduced in Eq. (8) is widely adopted by existing dimensionality reduction methods to eliminate redundancy among different projection directions and to prevent trivial scaling of the projection directions.

---

[2]Note that in most of scenarios, $\mathbf{S}_i$ is non-singular. To further ensure computational stability and robustness in extreme cases where $\mathbf{S}_i$ might become singular, we introduce a regularization term to the definition of the within-class-neighborhood scatter matrix: $\mathbf{S}_i = \sum_{\mathbf{x}_j \in \mathcal{N}_i \text{ or } \mathcal{N}_i^+} (\mathbf{x}_i - \mathbf{x}_j)(\mathbf{x}_i - \mathbf{x}_j)^T + \delta \mathbf{I}_D$. With a very small positive constant $\delta$, the term $\delta \mathbf{I}_D$ prevents potential singularities in $\mathbf{S}_i$.

## 3 Optimization

In this section, we present an approach to solve the optimization problem formulated in Eq. (8). Firstly, we convexify the constraint of the original problem. This transformation makes the problem explicitly solvable, facilitating a more tractable optimization process. Subsequently, we introduce an iterative optimization strategy to efficiently solve the transformed problem, with a guarantee of convergence.

The optimization problem in Eq. (8) is difficult to solve with respect to $\mathbf{W}$. To address this challenge, we reformulate the problem as follows:

$$\max_{\mathbf{Z}} \frac{\operatorname{tr}(\mathbf{\Sigma Z})}{\max_i \operatorname{tr}(\mathbf{S}_i \mathbf{Z})}, \tag{9}$$

where $\mathbf{Z} = \mathbf{W}\mathbf{W}^T$. The equivalence between the objective function in Eq. (8) and that in Eq. (9) can be easily established by using the cyclic property of the trace operator with proper matrix sizes, i.e., $\operatorname{tr}(\mathbf{ABC}) = \operatorname{tr}(\mathbf{BCA})$. To ensure consistency between the objective function and the constraint, we reformulate the constraint in Eq. (8), originally expressed in terms of $\mathbf{W}$, into an equivalent constraint set with respect to the new optimization variable $\mathbf{Z}$:

$$\Phi = \left\{ \mathbf{Z} \mid \mathbf{Z} = \mathbf{W}\mathbf{W}^T, \mathbf{W}^T\mathbf{W} = \mathbf{I}_d, \mathbf{W} \in \mathbb{R}^{D \times d} \right\}. \tag{10}$$

Given that $\Phi$ is non-convex, we introduce a convex relaxation by considering its convex hull (Overton & Womersley, 1993):

$$\Psi = \left\{ \mathbf{Z} \mid \operatorname{tr}(\mathbf{Z}) = d, \mathbf{0} \preceq \mathbf{Z} \preceq \mathbf{I}_D \right\}. \tag{11}$$

In fact, $\Psi$ is the smallest convex set containing $\Phi$, which approximates the original constraints to the greatest extent by making the relaxation as tight as possible, thereby reducing the likelihood that the relaxed optimizer falls outside $\Phi$. This relaxation allows us to replace the original constraint with a convex set, facilitating the application of standard optimization techniques while maintaining a close approximation to the original problem. Consequently, the optimization problem in Eq. (8) can be reformulated as follows:

$$\max_{\mathbf{Z}} \frac{\operatorname{tr}(\mathbf{\Sigma Z})}{\max_i \operatorname{tr}(\mathbf{S}_i \mathbf{Z})}, \quad \text{s.t. } \mathbf{Z} \in \Psi. \tag{12}$$

To solve this optimization problem, we employ an iterative strategy. Specifically, at the $t$-th iteration, we solve the following problem:

$$\mathbf{Z}^{(t)} = \arg\max_{\mathbf{Z}} \left\{ \operatorname{tr}(\mathbf{\Sigma Z}) - \alpha_t \max_i \operatorname{tr}(\mathbf{S}_i \mathbf{Z}) \right\}, \quad \text{s.t. } \mathbf{Z} \in \Psi, \tag{13}$$

where $\alpha_t = \frac{\operatorname{tr}(\mathbf{\Sigma Z}^{(t-1)})}{\max_i \operatorname{tr}(\mathbf{S}_i \mathbf{Z}^{(t-1)})}$ denotes the value of the objective function after the $(t-1)$-th iteration. Note that Eq. (13) follows the classical Dinkelbach-type parametric reformulation of fractional programming (Dinkelbach, 1967; Schaible, 1976), through which the original fractional problem in Eq. (12) can be equivalently solved by a sequence of parameterized subtractive problems. Solving Eq. (13) is equivalent to optimizing:

$$\mathbf{Z}^{(t)} = \arg\min_{\mathbf{Z}} \left\{ \alpha_t \max_i \operatorname{tr}(\mathbf{S}_i \mathbf{Z}) - \operatorname{tr}(\mathbf{\Sigma Z}) \right\}, \quad \text{s.t. } \mathbf{Z} \in \Psi, \tag{14}$$

which is obviously a convex problem, as $\max_i \operatorname{tr}(\mathbf{S}_i \mathbf{Z})$ is convex and $\alpha_t$ is a positive constant. As a result, it can be transformed into a standard semi-definite programming problem by introducing auxiliary variables $s$ and $u$:

$$\begin{aligned}
\min_{\mathbf{Z}, s, u} \quad & \alpha_t s - u \\
\text{s.t.} \quad & \operatorname{tr}(\mathbf{\Sigma Z}) \geq u > 0, \\
& \operatorname{tr}(\mathbf{S}_i \mathbf{Z}) \leq s, \forall i \\
& \operatorname{tr}(\mathbf{Z}) = d, \mathbf{0} \preceq \mathbf{Z} \preceq \mathbf{I}_D.
\end{aligned} \tag{15}$$

By solving this problem, we obtain the optimal $\mathbf{Z}^{(t)}$ at the $t$-th iteration, which is globally optimal for the Problem (13) and (14). We will repeat the above procedure until convergence. By the properties of Dinkelbach transform, the solution is globally optimal for the relaxed problem (12). After that, we can easily obtain the optimal $\mathbf{W}$ through the eigen-decomposition of $\mathbf{Z}$. Specifically, $\mathbf{W}$ is composed of eigenvectors corresponding to the largest $d$ eigenvalues of $\mathbf{Z}$. Such recovery allows that $\mathbf{W}$ is a feasible solution of the original problem since it satisfies the orthogonality constraint. The optimization procedure of NePuDA is summarized in Algorithm 1 in Appendix B.

## 4 Theoretical Analysis

In this section, we analyze the algorithmic convergence and computational complexity of the proposed method. The mathematical connections between the proposed method and existing methods are discussed in the Appendix C.

### 4.1 Convergence Analysis

To prove the convergence of the iterative procedure presented in Algorithm B1, we need to show both the monotonicity and boundedness of the optimization objective.

**Theorem 1.** *(Monotonicity) Let $h(\mathbf{Z}) = \frac{\mathrm{tr}(\mathbf{\Sigma Z})}{\max_i \mathrm{tr}(\mathbf{S}_i \mathbf{Z})}$, the value of $h(\mathbf{Z})$ is monotonically non-decreasing across successive iterations, i.e., $h(\mathbf{Z}^{(t)}) \geq h(\mathbf{Z}^{(t-1)})$.*

*Proof of Theorem* 1. We define $g(\mathbf{Z}) = \mathrm{tr}(\mathbf{\Sigma Z}) - \alpha_t \max_i \mathrm{tr}(\mathbf{S}_i \mathbf{Z})$. Since

$$\mathbf{Z}^{(t)} = \arg\max_{\mathbf{Z}} g(\mathbf{Z}) = \arg\max_{\mathbf{Z}} \left\{ \mathrm{tr}(\mathbf{\Sigma Z}) - \alpha_t \max_i \mathrm{tr}(\mathbf{S}_i \mathbf{Z}) \right\}, \tag{16}$$

we have

$$g(\mathbf{Z}^{(t)}) \geq g(\mathbf{Z}^{(t-1)}) = \mathrm{tr}(\mathbf{\Sigma Z}^{(t-1)}) - \alpha_t \max_i \mathrm{tr}(\mathbf{S}_i \mathbf{Z}^{(t-1)}) = 0. \tag{17}$$

The last equality is a direct consequence of the definition of $\alpha_t$, which is explicitly provided in the text immediately following Eq. (13). Hence, we have

$$g(\mathbf{Z}^{(t)}) = \mathrm{tr}(\mathbf{\Sigma Z}^{(t)}) - \alpha_t \max_i \mathrm{tr}(\mathbf{S}_i \mathbf{Z}^{(t)}) \geq 0$$

$$\Leftrightarrow \frac{\mathrm{tr}(\mathbf{\Sigma Z}^{(t)})}{\max_i \mathrm{tr}(\mathbf{S}_i \mathbf{Z}^{(t)})} \geq \alpha_t = h(\mathbf{Z}^{(t-1)}) \Leftrightarrow h(\mathbf{Z}^{(t)}) \geq h(\mathbf{Z}^{(t-1)}). \tag{18}$$

Thus, we conclude that $h(\mathbf{Z})$ is monotonically non-decreasing. $\qquad\square$

To show that $h(\mathbf{Z})$ is upper bounded, we first prove the following lemma.

**Lemma 1.** *For a symmetric and positive semi-definite matrix $\mathbf{A} \in \mathbb{R}^{D \times D}$ and a matrix $\mathbf{B} \in \Psi$, we have:*

$$\lambda_{\min}(\mathbf{A})\, d \leq \mathrm{tr}(\mathbf{AB}) \leq \lambda_{\max}(\mathbf{A}) d, \tag{19}$$

*where $\lambda(\cdot)$ denotes the eigenvalues of the given matrix.*

*Proof of Lemma* 1. Let $\mathbf{A} = \mathbf{U \Lambda U}^T$ be the eigen-decomposition of the matrix $\mathbf{A}$, where $\mathbf{\Lambda}$ is the diagonal eigenvalue matrix and $\mathbf{U}$ is the corresponding orthonormal eigenvector matrix. Accordingly, we have the following equality:

$$\mathrm{tr}(\mathbf{AB}) = \mathrm{tr}(\mathbf{BA}) = \mathrm{tr}(\mathbf{BU\Lambda U}^T) = \mathrm{tr}(\mathbf{\Lambda U}^T \mathbf{BU}) \tag{20}$$

We further assign a matrix $\hat{\mathbf{B}}$ as follows:

$$\left[\hat{\mathbf{B}}\right]_{i,j} = \begin{cases} \left[\mathbf{U}^T \mathbf{BU}\right]_{i,j}, & i = j, \\ 0, & i \neq j. \end{cases} \tag{21}$$

Then we have:

$$\mathrm{tr}\left(\boldsymbol{\Lambda}\mathbf{U}^T\mathbf{B}\mathbf{U}\right) = \mathrm{tr}\left(\boldsymbol{\Lambda}\hat{\mathbf{B}}\right) \leq \lambda_{\max}(\mathbf{A})\mathrm{tr}\left(\hat{\mathbf{B}}\right) = \lambda_{\max}(\mathbf{A})\mathrm{tr}\left(\mathbf{U}^T\mathbf{B}\mathbf{U}\right)$$

$$= \lambda_{\max}(\mathbf{A})\mathrm{tr}\left(\mathbf{B}\mathbf{U}\mathbf{U}^T\right) = \lambda_{\max}(\mathbf{A})\mathrm{tr}\left(\mathbf{B}\right) = \lambda_{\max}(\mathbf{A})d. \tag{22}$$

In Eq. (22), the first equality is derived from the diagonality of $\boldsymbol{\Lambda}$. By combining Eqs. (20) and (22), we know that the right-hand side inequality in Eq. (19) holds.

Similarly, for the left-hand side inequality in Eq. (19), we have the following:

$$\mathrm{tr}\left(\mathbf{A}\mathbf{B}\right) = \mathrm{tr}\left(\boldsymbol{\Lambda}\mathbf{U}^T\mathbf{B}\mathbf{U}\right) = \mathrm{tr}\left(\boldsymbol{\Lambda}\hat{\mathbf{B}}\right) \geq \lambda_{\min}(\mathbf{A})\mathrm{tr}\left(\mathbf{B}\right) = \lambda_{\min}(\mathbf{A})d.$$

With both Eqs. (22) and (23), we can conclude that the entire inequality in Eq. (19) holds. □

Based on the above Lemma 1, we can prove the boundness of the optimization objective in the following theorem.

**Theorem 2.** *(Boundness)* $\forall \mathbf{Z} \in \Psi$, $h(\mathbf{Z})$ *is upper bounded.*

*Proof of Theorem* 2. For the numerator of $h(\mathbf{Z})$, we have $\mathrm{tr}\left(\boldsymbol{\Sigma}\mathbf{Z}\right) \leq \lambda_{\max}(\boldsymbol{\Sigma})d$ according to Lemma 1, since $\boldsymbol{\Sigma}$ is symmetric and positive semi-definite. For the denominator of $h(\mathbf{Z})$, we have the following:

$$\max_i \mathrm{tr}(\mathbf{S}_i\mathbf{Z}) \geq \frac{1}{n}\sum_{i=1}^n \mathrm{tr}(\mathbf{S}_i\mathbf{Z}) \geq \frac{1}{n}\sum_{i=1}^n \lambda_{\min}(\mathbf{S}_i)d > 0.$$

Hence, we have

$$h(\mathbf{Z}) = \frac{\mathrm{tr}(\boldsymbol{\Sigma}\mathbf{Z})}{\max_i \mathrm{tr}(\mathbf{S}_i\mathbf{Z})} \leq \frac{\lambda_{\max}(\boldsymbol{\Sigma})}{\frac{1}{n}\sum_{i=1}^n \lambda_{\min}(\mathbf{S}_i)}. \tag{23}$$

□

Based on Theorems 1 and 2, we know that $h(\mathbf{Z})$ is monotonically non-decreasing and upper bounded, which guarantees the convergence of our algorithm's iterative procedure.

## 4.2 Complexity Analysis

The time/space complexity of our algorithm comprises four main components: nearest neighborhood construction, scatter matrix construction, SDP solver, and eigen-decomposition. The time and space complexities of the first component, nearest neighborhood construction, are $\Theta(n^2\log n)$ and $\Theta(nk)$, respectively. In terms of the construction of within-class-neighborhood scatter matrix $\mathbf{S}_i$, for NePuDA, the construction of $\mathbf{S}_i$ is $\mathcal{O}\left(nk\right)$, whereas that of NePuDA$^+$ is $\mathcal{O}\left(\sum_{i=1}^C {n^{(i)}}^2 \log n^{(i)}\right)$. For the construction of between-class-neighborhood scatter matrix $\boldsymbol{\Sigma}$, the time complexity is $\Theta(nk)$ and the space complexity is $\Theta(nD^2)$. For the proposed SDP procedure (15), its time complexity is $\Theta(\tau m_0^2 n_0^2)$, where $\tau$, $m_0$ and $n_0$ denote the time of iteration, the number of variables, and the size of the problem, respectively[3]. Finally, the time cost of the eigen-decomposition of $\mathbf{Z}$ to obtain $\mathbf{W}$ is $\Theta\left(D^3\right)$.

# 5 Experiments

In this section, we thoroughly validate the effectiveness of the proposed method across a range of supervised learning tasks using both synthetic and real-world datasets. We begin by presenting the baseline methods employed in our experiments. Next, we describe the experiments conducted on synthetic datasets, detailing the dataset characteristics and the corresponding experimental results. Following this, we discuss the experiments performed on real-world datasets, covering the dataset descriptions, experimental setup, result demonstration, and analysis.

---

[3]For more details of the complexity of SDP optimization, please refer to Appendix D.

## 5.1 Baseline Methods

We selected 14 representative linear dimensionality reduction methods for performance comparison, comprising 2 unsupervised methods, PCA (Hotelling, 1933) and FSPCA (Nie et al., 2023), and 12 supervised methods: the original LDA (Fisher, 1936); FastSDA (Chumachenko et al., 2021), RSLDA (Wang et al., 2022a), TRLDA (Wang et al., 2022b), RLDA (Zhao et al., 2019), and $\ell_{2,1}$-LDA (Nie et al., 2021) from the metric modification category; WLDA (Zhang & Yeung, 2010), HMMDA (Su et al., 2018) and WDDR (Wang et al., 2024b) from the max-min strategy category; ANMMP (Zhao et al., 2018), LMNN (Weinberger & Saul, 2009), and NCA (Goldberger et al., 2004) from the neighborhood exploration category. More details on these baseline methods are provided in the Appendix E.

## 5.2 Experiments on Synthetic Dataset

In this section, we introduce two synthetic datasets designed to evaluate the effectiveness of our proposed method. The first dataset (detailed in Section 5.2.1)illustrates the discriminative power of the subspace identified by our approach, i.e., how well it separates classes. The second dataset (presented in Section 5.2.2)focuses on verifying the neighborhood purification effect within the subspace, showing how effectively the proposed method cleans up local neighborhoods. These constructed synthetic scenarios allow us to highlight the key strengths of our method in a controlled setting before moving to real-world data.

### 5.2.1 Case Study 1: Elliptical Distribution

In this subsection, we use a synthetic dataset to illustrate the effectiveness of the proposed idea in identifying the discriminative subspace. The dataset is composed of 300 two-dimensional data samples from three classes, with 100 data points per class. Each class is represented by a narrow elliptical distribution, with the length of the major axis being 20 and that of the minor axis being 3, as illustrated in Fig. 3. To assess the robustness of the proposed method, we add uniformly distributed noise in the range of $[0, 1]$ along the vertical axis of each data sample. The objective of this learning task is to identify a one-dimensional subspace in which the three classes are well separated.

In this dataset, the two classes at the top (i.e., the red class and the green class) are close to each other, presenting challenges for the supervised dimensionality reduction task. Additionally, the direction of the major axis of the bottom class (i.e., the blue class) is orthogonal to that of the top classes. This further complicates the task of finding a projection direction that minimizes the within-class scatter for all classes, thereby bringing additional challenges to the subspace learning task.

We compare the proposed methods (NePuDA and NePuDA$^+$) with 6 baseline methods from all three main categories summarized in Section 1 and Appendix A.3. They are: FSPCA, LDA, FastSDA, RSLDA, WLDA, and ANMMP. The projection results of all methods are presented in Fig. 3. The FSPCA (Fig. 3a), being unsupervised, selects a direction that maximizes the overall variance; however, this may not be optimal for discriminative analysis. Traditional LDA (Fig. 3b) and its metric modification variants (Fig. 3c and Fig. 3d) focus on maximizing the overall between-class scatter and minimizing the within-class scatter, which can lead to overlaps between nearby classes in the learned subspace. For the WLDA (Fig. 3e), although it takes special care of closely situated classes, it operates at the class level, using the mean of each class to represent the entire class. This can overlook individual data points, resulting in overlaps between different classes in the low-dimensional projections. The proposed method (both versions under various parameter settings, as shown in Figs. 3g-3l) and the ANMMP (Fig. 3f), which emphasize the local distribution of data and aim to ensure the separation of nearby data points from different classes, achieve better projections results compared to the other five methods. Furthermore, the proposed method consistently produces clearly separable projections across all settings, demonstrating its robustness with respect to variations in both the version of the method and the neighborhood size.

### 5.2.2 Case Study 2: Partial Spiral Distribution

In this subsection, we introduce another synthetic dataset specifically designed to showcase NePuDA's ability to purify local neighborhoods, even in situations where global class separation is infeasible. The dataset

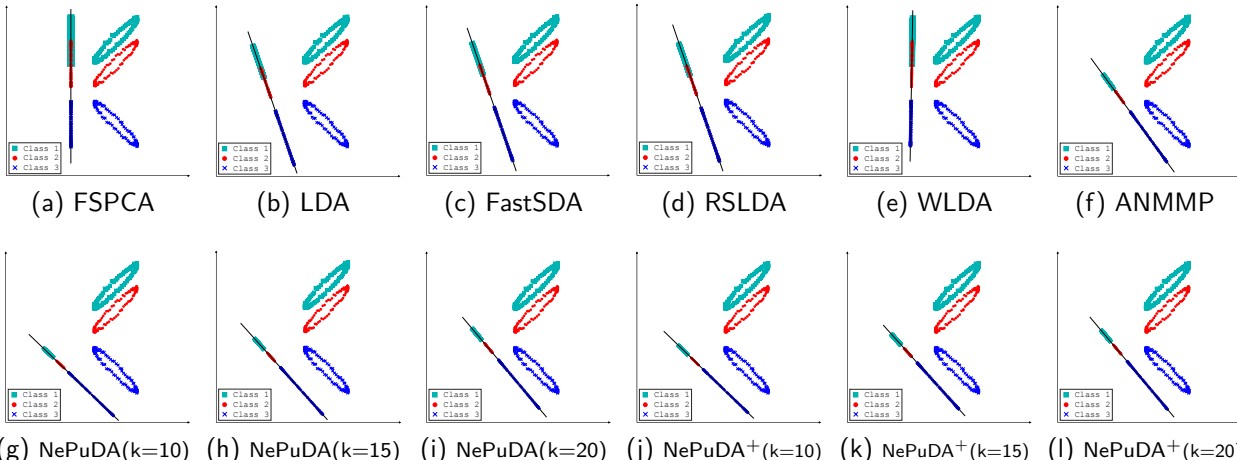

(a) FSPCA     (b) LDA     (c) FastSDA     (d) RSLDA     (e) WLDA     (f) ANMMP

(g) NePuDA(k=10)   (h) NePuDA(k=15)   (i) NePuDA(k=20)   (j) NePuDA$^+$(k=10)   (k) NePuDA$^+$(k=15)   (l) NePuDA$^+$(k=20)

Figure 3: Performance comparison of FSPCA (Nie et al., 2023), LDA (Fisher, 1936), FastSDA (Chumachenko et al., 2021), RSLDA (Wang et al., 2022a), WLDA (Zhang & Yeung, 2010), ANMMP (Zhao et al., 2018), and two versions of the proposed method, NePuDA and NePuDA$^+$, on a two-dimensional synthetic dataset.

consists of points lying on a spiral manifold in 2D space, parameterized by the following polar equations:

$$\begin{cases} x = 0.2\theta \cos\theta, \\ y = 0.2\theta \sin\theta, \end{cases} \tag{24}$$

with $\theta \geq 0$. We assign two class labels (represented by green crosses and red triangles, respectively) in an interleaved manner along the spiral arms. Unlike the elliptical example in Section 5.2.1, no linear projection can fully separate the two classes across the entire dataset. Our goal here is therefore more modest yet practically meaningful: to find a one-dimensional subspace in which each data point is surrounded by same-class neighbors in the projected space. Such a subspace should provide sufficiently clean local structure to enable reliable classification using a simple nearest neighbor rule.

We compare NePuDA with 6 representative baselines: FSPCA, LDA, RSLDA, TRLDA, WLDA, and AN-MMP. Projection results of all methods are presented in Fig. 4, with a dashed box in each subfigure magnifying a representative region for closer inspection. Methods that operate primarily at the class or subclass level (FSPCA, LDA, RSLDA, TRLDA, WLDA) produce subspaces with noticeable class overlaps in the subspace, as shown in Figs. 4a to 4e. Even ANMMP, which explicitly considers neighborhood information, fails to ensure pure neighborhoods for all points; while overlap is reduced, small but persistent mixing remains visible in Fig. 4f. By contrast, NePuDA successfully achieves clean neighborhood purification across the dataset. As shown in Fig. 4g and Fig. 4h, points in the projected subspace are consistently surrounded by same-class neighbors, with virtually no local contamination.

## 5.3 Experiments on Real-world Datasets

In this subsection, we present a comprehensive comparative analysis utilizing 13 real-world datasets. The structure of this section is organized as follows: Section 5.3.1 provides a brief overview of the datasets used in our experiments. Section 5.3.2 describes the experimental setup. Section 5.3.3 presents the experimental results in tables, reporting the best performance and the corresponding dimensionality achieved by each method. Section 5.3.4 offers a case study using the JAFFE dataset to visually demonstrate the effectiveness of our proposed method. We compare the projections generated by our approach against those of baseline methods, providing an intuitive understanding of the improvements achieved. In addition to experiments discussed in the main paper, we further extend our proposed methods on 3 additional large-scale datasets and explore their integration with deep pre-trained models and nonlinear evaluations in Appendix I. We also empirically explore the sensitivity of our method to various parameter configurations and validate the convergence of the algorithm, which are presented in Appendices F and G, respectively.

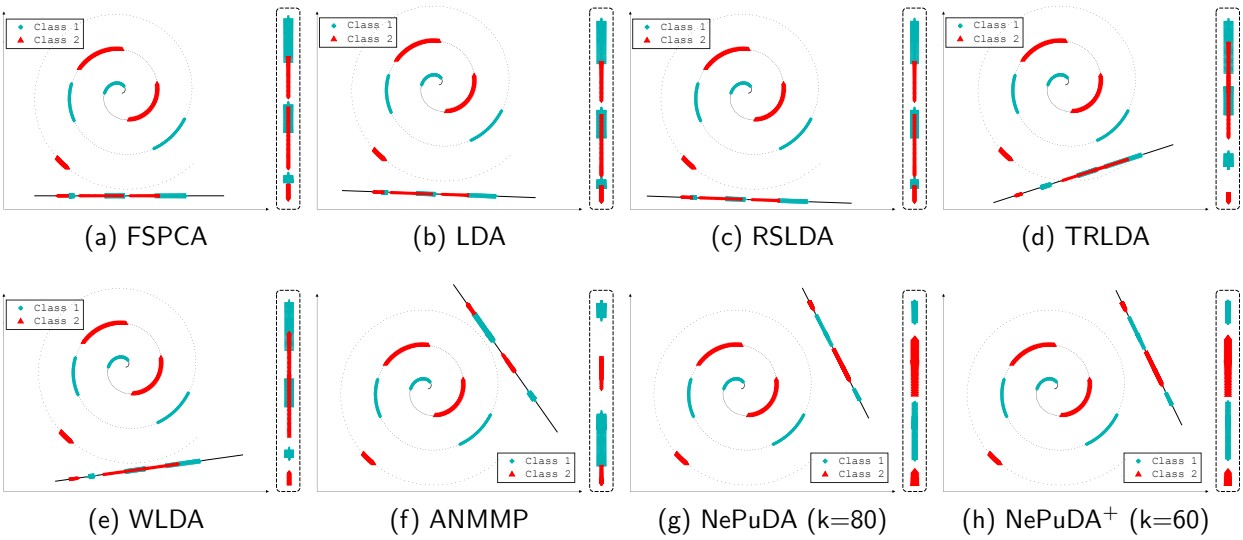

Figure 4: Performance comparison of FSPCA (Nie et al., 2023), LDA (Fisher, 1936), RSLDA (Wang et al., 2022a), TRLDA (Wang et al., 2022b), WLDA (Zhang & Yeung, 2010), ANMMP (Zhao et al., 2018), and two versions of the proposed method, NePuDA and NePuDA$^+$, on the two-dimensional synthetic dataset.

### 5.3.1 Dataset Description

Among the 13 datasets used in our empirical evaluation, Iris is one of the most classical datasets for classification tasks, while Spiral is a small 2D dataset. We also selected various UCI datasets from different domains, including social science (e.g., Hayes-Roth and Balance-scale), movement and gesture (e.g., Libras), image segmentation (e.g., Segment), physics and chemistry (e.g., Glass, Sonar, and Ionosphere), and biology (e.g., Yeast). Additionally, we incorporated human-face datasets to validate our method, as these datasets feature high-dimensional data with intertwined classes in the original space, making them ideal for assessing the effectiveness of supervised dimensionality reduction methods. The human-face datasets used include JAFFE, Yale, and the Extended Yale Face Database B (YaleB). JAFFE is a facial expression dataset containing 7 expressions from 10 individuals. The Yale dataset consists of 165 grayscale facial images of 15 different individuals, while YaleB is an extension of the Yale dataset, providing more facial images and greater variations. In addition to the datasets evaluated in the main paper, we include three large-scale datasets, Isolet, MNIST, and FashionMNIST, in Appendix I, to provide a more comprehensive evaluation of the proposed methods in large-scale settings, including their integration with deep architectures and evaluation using nonlinear classifiers. Please refer to Appendix H for a brief summary of the key statistics (size, original dimensionality, and number of classes) of these datasets.

### 5.3.2 Experimental Setup

We employ dataset-specific preprocessing strategies for human-face datasets and non-human-face datasets. For non-human-face datasets, we generally input the raw data directly into our model and 14 baseline models without preliminary dimensionality reduction. However, an exception is made for the Libras dataset due to its high redundancy. In this case, we perform PCA and select top 20 principal components, which preserve 99.83% of the total energy. For all human-face datasets, we utilize PCA to reduce the original dimensionality while preserving at least 90% of the total variance (energy) in the data. These PCA-transformed data then serve as the input for all 13 models under evaluation. Specifically, for the JAFFE dataset, PCA reduces the original feature space to 60 dimensions, preserving 92.56% of the total variance. The dimension of the original Yale dataset is reduced from 1024 to 50, retaining 91.72% of the variance. For the YaleB dataset, PCA compresses the original 1024-dimensional space to 35 dimensions, maintaining 92.45% of the total variance.

For the parameter setting, we employ distinct strategies for the two versions of our proposed method to set the neighborhood size. Specifically, for NePuDA, it is selected from the set: $\{\lfloor \min_{j=1,\ldots,C} n^{(j)}/2 \rfloor \times i,\ i \in \{1,2,3,4\}\}$, where $n^{(j)}$ represents the sample size of the $j$-th class. For NePuDA$^+$, it is selected from: $\{\lfloor \min_{j=1,\ldots,C} n^{(j)}/2^i \rfloor,\ i \in \mathbb{Z}^+ \ \& \ i \leq \log_2 \min_{j=1,\ldots,C} n^{(j)} - 1\}$. The parameter is determined by 5-fold cross validation, to avoid any potential bias. The difference in the ranges of the two proposed methods arises from their distinct mechanisms for constructing pure neighborhoods. Since NePuDA first selects neighboring data points and then picks same-class data points, a relatively larger $k$ is needed to ensure sufficient within-class neighbors are considered. In contrast, NePuDA$^+$ starts from same-class data points and then considers neighboring data points, so a smaller $k$ is generally sufficient to build up the within-class-neighborhood scatter. These parameter ranges are selected based on the observed robustness of our methods to parameter variations, which will be discussed in subsequent sections. We set $\epsilon = 10^{-4}$ as the stopping criterion. For the optimization toolbox, we use the default precision and maximum iteration settings provided by CVX with SDPT3 in MATLAB. The initial $\mathbf{Z}^{(0)}$ is randomly initialized with entries drawn independently from a uniform distribution over $(0,1)$, serving as the starting point of the iterative optimization procedure. Regarding the implementation of comparison methods, we utilize a combination of existing resources and custom implementations. For PCA, we employ MATLAB's built-in function. For LDA, worst-case LDA, HMMDA, WDDR, LMNN, and NCA, we implement them according to the techniques presented in the original paper. For FSPCA, FastSDA, ANMMP, RLDA, $\ell_{2,1}$-LDA, RSLDA, and TRLDA, we use the provided source code, with parameters for ANMMP, RSLDA, and TRLDA set in accordance with their original publications.

For each dataset and dimensionality, we conduct five independent trials, and calculate the average classification accuracy across these five trials. We record the dimensionality that yields the highest average accuracy, along with its corresponding standard deviation. In each trial, we randomly select around 70% of the samples from each class for training, with the remaining samples serving as the test set. This split ratio aligns with established practices in the literature (Zhang & Yeung, 2010; Weinberger & Saul, 2009). Following common practice in recent work on dimensionality reduction, we evaluate classification performance in the learned subspace using a straightforward $k$-Nearest Neighbors ($k$NN) classifier (Wang et al., 2024b; Omati et al., 2025) with $k = 3$ (Wang et al., 2022a;b).

### 5.3.3 Classification Performance Comparison

The experimental results are presented in Tables 1 and 2, which provide a comprehensive overview of all methods' performance across various datasets. Table 1 showcases the testing results for relatively low-dimensional datasets, while Table 2 gives the results for higher-dimensional datasets. For each dataset, we highlight the highest achieved accuracy in bold, accompanied by its corresponding standard deviation. Additionally, we report the dimensionality of the subspace that yields this optimal accuracy for each method, denoted as 'r.d.' (reduced dimension) in our tables.

In Table 1, we can observe that NePuDA achieves the highest accuracy on the Yeast dataset, outperforming all comparison methods. Notably, NePuDA$^+$ consistently performs the best on multiple datasets, including Iris, Hayes-roth, Glass, and Balance-scale. While TRLDA marginally outperforms our methods on the Spiral dataset, the difference is not large. Moreover, the low standard deviations and the low dimensionality of the learned subspace across datasets demonstrate the stability of the proposed method and its ability to extract discriminative information.

In Table 2, NePuDA is the best among all the comparison methods on the Sonar dataset, while NePuDA$^+$ achieves the highest accuracy on Segment, Libras, and all human-face datasets. The only minor exception is the Ionosphere dataset, where NePuDA$^+$ is slightly outperformed by $\ell_{2,1}$-LDA and WDDR, but still maintains competitive performance. The consistent superiority of NePuDA$^+$ on human face datasets is noteworthy, given the challenging nature of these datasets, where the intra-class variance can exceed the inter-class distance. For instance, in the JAFFE dataset, which involves emotion classification, two facial images of different individuals with the same emotion (considered as the same class) typically exhibit a much larger Euclidean distance than two images of the same person with different emotions (considered as different classes), presenting great challenges to supervised dimensionality reduction. The proposed method, NePuDA, prioritizes preserving the purity within the neighborhood of each individual data sample, rather

Table 1: Performance comparison in terms of classification accuracy and standard deviation for 14 baseline approaches and two versions of the proposed method on 6 relatively low-dimensional datasets. Here, *r.d.* denotes the reduced dimension that yields the highest accuracy for each method, and *Accuracy* represents the mean accuracy over 5 trials.

| methods | metric | Iris | Spiral($2D$) | Hayes-roth | Yeast | Glass | Balance-scale |
|---|---|---|---|---|---|---|---|
| PCA | Accuracy | 93.33±4.16 | 45.38±6.20 | 43.59±4.05 | 53.02±2.75 | 39.38±17.86 | 56.94±11.69 |
| | r.d. | $d=3$ | $d=1$ | $d=3$ | $d=7$ | $d=6$ | $d=3$ |
| FSPCA | Accuracy | 94.67±3.37 | 52.90±6.51 | 45.64±7.78 | 53.92±2.47 | 65.23±5.92 | 70.71±2.64 |
| | r.d. | $d=2$ | $d=1$ | $d=5$ | $d=7$ | $d=6$ | $d=3$ |
| LDA | Accuracy | 95.11±0.99 | 50.97±1.23 | 65.64±4.66 | 52.70±0.83 | 62.15±6.31 | 84.59±3.71 |
| | r.d. | $d=2$ | $d=1$ | $d=2$ | $d=4$ | $d=3$ | $d=3$ |
| FastSDA | Accuracy | 96.00±1.86 | 57.42±1.80 | 67.69±8.81 | 54.77±2.73 | 68.92±4.13 | 85.88±2.39 |
| | r.d. | $d=2$ | $d=1$ | $d=2$ | $d=6$ | $d=4$ | $d=3$ |
| RSLDA | Accuracy | 96.00±2.43 | 53.12±2.70 | 70.26±10.35 | 55.68±3.21 | 68.62±2.57 | 87.18±2.74 |
| | r.d. | $d=2$ | $d=1$ | $d=2$ | $d=7$ | $d=6$ | $d=3$ |
| TRLDA | Accuracy | 95.11±4.27 | **57.42±4.27** | 62.00±10.95 | 54.19±1.73 | 66.77±4.43 | 87.88±1.69 |
| | r.d. | $d=3$ | $d=1$ | $d=3$ | $d=7$ | $d=7$ | $d=3$ |
| RLDA | Accuracy | 94.67±2.53 | 54.41±8.00 | 68.21±4.29 | 51.35±1.80 | 66.46±7.25 | 88.12±3.75 |
| | r.d. | $d=2$ | $d=1$ | $d=1$ | $d=7$ | $d=8$ | $d=2$ |
| $\ell_{2,1}$-LDA | Accuracy | 95.56±1.57 | 56.13±4.65 | 69.23±5.13 | 51.89±1.40 | 66.77±6.21 | 88.00±1.15 |
| | r.d. | $d=3$ | $d=1$ | $d=4$ | $d=7$ | $d=8$ | $d=1$ |
| WLDA | Accuracy | 96.00±0.99 | 53.55±2.98 | 69.23±7.48 | 52.03±2.75 | 69.54±5.48 | 86.71±2.84 |
| | r.d. | $d=3$ | $d=1$ | $d=1$ | $d=7$ | $d=7$ | $d=2$ |
| HMMDA | Accuracy | 94.22±3.72 | 52.90±3.35 | 71.28±10.48 | 53.54±2.75 | **70.77±4.41** | 87.65±1.32 |
| | r.d. | $d=1$ | $d=1$ | $d=2$ | $d=7$ | $d=2$ | $d=2$ |
| WDDR | Accuracy | 95.56±3.14 | 53.98±3.98 | 71.79±13.20 | 53.83±1.36 | 67.08±5.82 | 87.84±2.72 |
| | r.d. | $d=1$ | $d=1$ | $d=3$ | $d=7$ | $d=9$ | $d=2$ |
| ANMMP | Accuracy | 96.00±2.90 | 54.84±1.70 | 71.28±4.93 | 54.10±1.12 | 69.85±4.69 | 87.06±0.83 |
| | r.d. | $d=2$ | $d=1$ | $d=1$ | $d=6$ | $d=8$ | $d=1$ |
| NCA | Accuracy | 92.44±2.43 | 52.81±5.84 | 68.72±2.92 | 49.23±1.84 | 67.50±6.83 | 89.15±2.81 |
| | r.d. | $d=2$ | $d=1$ | $d=4$ | $d=6$ | $d=8$ | $d=2$ |
| LMNN | Accuracy | 96.44±1.22 | 50.00±4.47 | 68.21±6.88 | 54.82±2.85 | 66.87±2.84 | 81.90±3.91 |
| | r.d. | $d=1$ | $d=1$ | $d=2$ | $d=7$ | $d=4$ | $d=2$ |
| NePuDA | Accuracy | 97.04±1.28 | 55.91±4.30 | 62.05±4.21 | **56.22±0.95** | 66.67±7.11 | 76.47±8.68 |
| | r.d. | $d=1$ | $d=1$ | $d=1$ | $d=7$ | $d=8$ | $d=2$ |
| NePuDA$^+$ | Accuracy | **98.52±1.28** | 56.63±6.91 | **72.82±11.15** | 54.28±0.87 | 70.46±4.13 | **89.65±2.68** |
| | r.d. | $d=3$ | $d=1$ | $d=2$ | $d=7$ | $d=8$ | $d=2$ |

than pursuing the impractically high standard of identifying an ideal, whole-class-separated subspace, thus resulting in better performance compared to other baseline methods. We will discuss more details of the results on the JAFFE dataset as a case study in the next subsection. Meanwhile, ANMMP serves as the strongest baseline in terms of average accuracy among all 13 datasets, and we conduct paired $t$-tests with NePuDA$^+$ and ANMMP in terms of mean accuracy across datasets. Statistically, NePuDA$^+$ significantly outperforms ANMMP (mean improvement = 1.95%, p-value = 0.0043) at the 1% significance level.

Table 2: Performance comparison in terms of classification accuracy and standard deviation for 14 baseline approaches and two versions of the proposed method on 7 relatively high-dimensional datasets. Here, *r.d.* denotes the reduced dimension that yields the highest accuracy for each method, and *Accuracy* represents the mean accuracy over 5 trials.

| methods | metric | Sonar | Segment | Libras | Ionosphere | JAFFE | Yale | YaleB |
|---|---|---|---|---|---|---|---|---|
| PCA | Accuracy | 72.58±4.84 | 77.78±4.20 | 64.57±9.37 | 77.90±6.08 | 25.52±0.90 | 34.81±8.98 | 24.94±10.92 |
| | r.d. | $d=21$ | $d=4$ | $d=11$ | $d=10$ | $d=17$ | $d=13$ | $d=29$ |
| FSPCA | Accuracy | 83.33±2.46 | 78.31±3.67 | 78.48±2.74 | 88.38±3.04 | 50.52±3.25 | 65.19±7.14 | 44.65±1.34 |
| | r.d. | $d=28$ | $d=7$ | $d=17$ | $d=3$ | $d=39$ | $d=9$ | $d=33$ |
| LDA | Accuracy | 75.27±5.66 | 84.12±3.93 | 79.05±1.65 | 84.23±6.06 | 73.96±2.39 | 79.26±1.28 | 80.48±1.28 |
| | r.d. | $d=1$ | $d=5$ | $d=11$ | $d=1$ | $d=5$ | $d=12$ | $d=18$ |
| FastSDA | Accuracy | 73.12±3.36 | 86.77±3.67 | 80.00±1.90 | 83.62±2.17 | 75.52±2.39 | 81.48±7.80 | 83.87±2.23 |
| | r.d. | $d=37$ | $d=4$ | $d=12$ | $d=1$ | $d=6$ | $d=13$ | $d=22$ |
| RSLDA | Accuracy | 81.29±2.93 | 82.54±5.72 | 78.41±1.45 | 88.95±2.57 | 85.94±2.71 | 85.33±6.40 | 85.80±1.08 |
| | r.d. | $d=30$ | $d=8$ | $d=18$ | $d=9$ | $d=8$ | $d=19$ | $d=33$ |
| TRLDA | Accuracy | 80.37±1.89 | 77.78±5.72 | 77.71±4.40 | 89.33±3.53 | 67.19±5.49 | 68.15±4.63 | 85.43±1.45 |
| | r.d. | $d=30$ | $d=13$ | $d=12$ | $d=6$ | $d=14$ | $d=25$ | $d=33$ |
| RLDA | Accuracy | 79.57±5.66 | 89.42±5.10 | 80.95±3.43 | 87.34±3.76 | 76.04±5.49 | 83.70±6.79 | 84.75±0.70 |
| | r.d. | $d=47$ | $d=15$ | $d=8$ | $d=4$ | $d=12$ | $d=6$ | $d=33$ |
| $\ell_{2,1}$-LDA | Accuracy | 83.33±1.86 | 84.66±3.30 | 80.00±2.86 | **90.95±0.67** | 79.69±2.71 | 86.67±7.80 | 85.21±0.56 |
| | r.d. | $d=52$ | $d=7$ | $d=17$ | $d=10$ | $d=13$ | $d=9$ | $d=21$ |
| WLDA | Accuracy | 82.26±5.59 | 89.42±3.67 | 76.38±5.19 | 89.33±2.81 | 80.21±5.92 | 85.19±5.59 | 84.71±1.62 |
| | r.d. | $d=24$ | $d=9$ | $d=13$ | $d=15$ | $d=15$ | $d=11$ | $d=26$ |
| HMMDA | Accuracy | 80.11±3.72 | 89.42±3.76 | 79.37±1.45 | 87.34±2.77 | 77.95±3.46 | 83.34±4.46 | 86.97±1.49 |
| | r.d. | $d=30$ | $d=13$ | $d=17$ | $d=4$ | $d=11$ | $d=12$ | $d=28$ |
| WDDR | Accuracy | 83.23±5.42 | 84.92±1.12 | 80.19±4.83 | 90.86±3.79 | 73.44±5.47 | 84.44±8.40 | 85.12±2.57 |
| | r.d. | $d=54$ | $d=18$ | $d=11$ | $d=9$ | $d=5$ | $d=30$ | $d=29$ |
| ANMMP | Accuracy | 83.87±1.61 | 84.76±2.88 | 80.38±3.73 | 89.90±2.90 | 82.29±1.80 | 84.44±3.85 | 86.98±0.71 |
| | r.d. | $d=50$ | $d=16$ | $d=12$ | $d=6$ | $d=12$ | $d=21$ | $d=27$ |
| NCA | Accuracy | 80.65±5.10 | 85.71±1.80 | 77.22±1.97 | 87.55±2.43 | 82.31±5.01 | **86.80±3.28** | 78.44±2.17 |
| | r.d. | $d=19$ | $d=14$ | $d=17$ | $d=12$ | $d=22$ | $d=49$ | $d=23$ |
| LMNN | Accuracy | 84.84±3.88 | 86.35±2.31 | 78.90±4.97 | 86.79±3.26 | 83.88±4.69 | 86.00±2.75 | 85.33±0.42 |
| | r.d. | $d=51$ | $d=17$ | $d=7$ | $d=10$ | $d=52$ | $d=42$ | $d=30$ |
| NePuDA | Accuracy | **85.48±4.27** | 85.71±6.73 | 80.19±3.12 | 89.21±2.40 | 83.33±7.86 | 82.22±1.28 | 83.45±0.61 |
| | r.d. | $d=47$ | $d=15$ | $d=14$ | $d=9$ | $d=7$ | $d=22$ | $d=24$ |
| NePuDA$^+$ | Accuracy | 83.87±4.27 | **90.48±2.24** | **81.43±2.52** | 90.67±1.56 | **88.54±1.80** | 86.67±2.42 | **87.07±0.74** |
| | r.d. | $d=46$ | $d=12$ | $d=15$ | $d=13$ | $d=16$ | $d=9$ | $d=17$ |

Besides, we can observe that NePuDA$^+$ achieves generally better results than NePuDA. A possible reason is that NePuDA relies on same-label samples within the original geometric neighborhoods, which can be effective when local regions contain sufficient same-class samples, like the case of Yeast and Sonar. However, when the original neighborhoods are highly mixed with relatively limited same-class samples, such pure-neighborhood structures may become insufficient or less stable. NePuDA$^+$ alleviates this issue by directly constructing same-class neighborhoods, which provides a more stable and informative same-class local set for

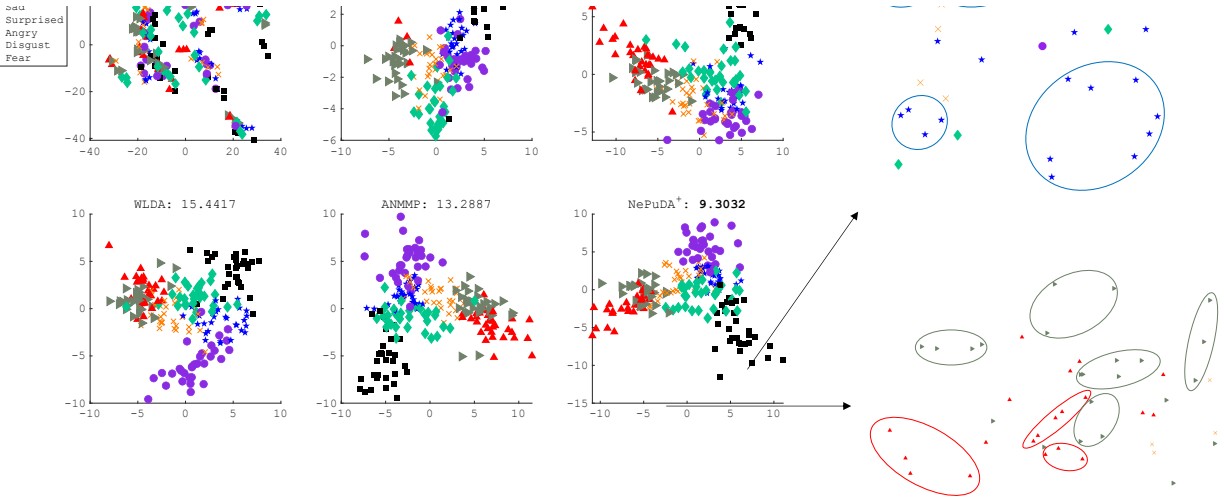

Figure 5: Visualization of the projection results of FSPCA, LDA, RSLDA, WLDA, ANMMP, and NePuDA$^+$ on JAFFE dataset. For each method, the projection result is selected from the reduced dimension that yields the highest classification accuracy.

each sample, thereby achieving a stronger overall performance. This demonstrates that an appropriate pure-neighborhood construction for specific data distribution is important, while the shared min-max objective remains the core mechanism for enforcing neighborhood purification.

### 5.3.4 Result Visualization on JAFFE Dataset

In this subsection, we further demonstrate the effectiveness of the proposed method by visualizing the projection results on the JAFFE dataset, which exhibits large intra-class distances (i.e., different individuals expressing the same emotion) and small inter-class distances (i.e., the same individual expressing different emotions), as shown in Appendix J. These characteristics make it particularly challenging to achieve effective whole-class separation in the projected subspace. Our visualization aims to illustrate how our method addresses these challenges by focusing on local neighborhood refinement rather than global class separation.

We visualize the projection results of six methods: the unsupervised FSPCA, the classical LDA, the RSLDA from the metric modification category, the WLDA from the max-min strategy category, the ANMMP from the neighborhood exploration category, and the proposed NePuDA$^+$. For each method, we utilize the projection in the subspace that yields the highest classification accuracy. To facilitate visual comparison, we further project these results onto a 2-D space using PCA. We employ distinct colors and markers to represent different classes (i.e., emotions).

The visualization of the projection results for all methods is presented in Fig. 5. As expected, the five supervised DR methods achieve more separable projections compared to the unsupervised FSPCA method, aligning with their learning objective of maximizing the separability in the subspace. Among the supervised DR methods, we observe consistent patterns in class distributions. The neutral class (blue stars) exhibits proximity to, and some overlap with, the happy and fear classes (purple circles and green diamonds, respectively). Similarly, the sad, angry, and disgust classes (orange crosses, red triangles, and gray triangles, respectively) demonstrate close proximity or overlap with each other.

To quantify the effectiveness of each method in minimizing class overlap and enhancing separation, we employ a metric based on the overlapping areas between classes in the 2-D space. For each class, we frame it using a minimum bounding rectangle and calculate the area of overlap between these rectangles for each class pair. We then compute the arithmetic mean of these overlapping areas across all class pairs for each method. These mean overlap values are presented at the top of each subfigure, alongside the method name.

The results show that our proposed method, NePuDA$^+$, achieves the lowest mean overlap between classes. This quantitative measure provides empirical support for the superior performance of our method in this challenging task, where clear class separation is difficult due to the complex nature of facial emotion data.

To further analyze the effectiveness of the proposed idea of neighborhood purification, we zoom in and examine two overlapping areas in the NePuDA$^+$ projection result, as shown on the right-hand side of Fig. 5. Our method aims to purify the neighborhood structure around individual data points as much as possible. As a result, even in overlapping regions, most points are surrounded by samples from their own class. This desirable pure neighborhood facilitates accurate classification, even with a simple nearest neighbor classifier, which is particularly useful in scenarios with complex intra-class variations and inter-class similarities.

## 6    Conclusions

In this paper, we introduced a novel supervised dimensionality reduction method called Neighborhood-Purifying Discriminant Analysis (NePuDA). By uncovering the subspace that encourages data samples to be purely surrounded by neighbors from the same class, the proposed method demonstrated competitive performance on both synthetic and 13 real-world datasets when compared with 14 representative baselines in dimensionality reduction. Our theoretical analyses provided support for NePuDA's superior performance.

**Limitations and Future Directions.** Despite these promising results, the current formulation has two main limitations. First, because NePuDA relies on semidefinite programming (SDP) to solve the core optimization problem (as analyzed in Section 4.2), its computational cost grows significantly when both the number of samples $n$ and the original dimensionality $D$ are large, making it challenging to scale to very big or extremely-high dimensional datasets. Second, the method is designed as a standalone dimensionality reduction step and remains agnostic to the downstream classifier; this modular approach ensures fair evaluation of the learned subsapce itself but may limit gains in full end-to-end pipelines. In the future, we plan to pursue three main directions to address these issues and broaden the applicability of the proposed method. First, we will explore faster optimization alternatives to SDP that can handle larger $n$ and $D$ without sacrificing solution quality. Promising candidates include reformulations based on the constrained concave-convex procedure (CCCP), second-order cone programming (SOCP), or carefully designed dual problems that enable more efficient solvers. Second, while the current comparisons focus on standalone DR methods to isolate subspace discriminativeness, we intend to develop an integrated, end-to-end framework that couples NePuDA-style subspace learning directly with classification. This will allow benchmarking against strong supervised baselines such as SVMs and modern deep neural networks. For example, one possible direction is to incorporate our objective into deep architectures as a differentiable subspace-learning module or auxiliary loss, thereby forming a unified end-to-end deep model and establishing a more coherent integration with representation learning. Last but not least, to tackle nonlinear data more effectively, we plan to extend NePuDA using kernel methods and deep architectures, while preserving, or ideally strengthening, its theoretical guarantees and maintaining reasonable computational efficiency.

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

## A    Notations and Brief Review of Related Work

### A.1    Notations and Descriptions

In this subsection, we summarize the notations and other frequently-used symbols in Table. 3.

Table 3: Notations and descriptions.

| Notations | Descriptions |
|:---:|:---:|
| $\mathbf{X}$ | The data matrix |
| $n$ | The total number of data samples |
| $\boldsymbol{\mu}$ | The mean vector of all data samples |
| $\mathbf{x}_i$ | The $i$-th data sample |
| $\ell_i$ | The label of $\mathbf{x}_i$ |
| $D$ | The dimensionality of original feature space |
| $d$ | The reduced dimensionality of the subspace |
| $C$ | The number of classes |
| $\mathbf{X}^{(j)}$ | The data matrix of the $j$-th class |
| $n^{(j)}$ | The number of data samples in the $j$-th class |
| $\boldsymbol{\mu}^{(j)}$ | The mean vector of the $j$-th class |
| $\mathbf{S}_w$ | The within-class scatter matrix |
| $\mathbf{S}_b$ | The between-class scatter matrix |
| $\mathbf{W}$ | The projection matrix |
| $\mathbf{w}_m$ | The $m$-th column of the projection matrix $\mathbf{W}$ |
| $\mathbb{N}_k(\Omega, \mathbf{x}_i)$ | The neighborhood set of $\mathbf{x}_i$ within the set $\Omega$ with size $k$ |
| $|\Omega|$ | The size or cardinality of $\Omega$ |
| $\mathcal{N}_i/\mathcal{N}_i^+$ | The pure neighborhood (within-class neighborhood) of $\mathbf{x}_i$ |
| $\mathcal{N}_i^c$ | Between-class neighborhood of $\mathbf{x}_i$ |
| $\mathbf{S}_i$ | The within-class-neighborhood scatter matrix of $\mathbf{x}_i$ |
| $\boldsymbol{\Sigma}_i$ | The between-class-neighborhood scatter matrix of $\mathbf{x}_i$ |

### A.2    Brief Review of LDA

Let $\mathbf{X} = \begin{bmatrix} \mathbf{x}_1, \mathbf{x}_2, \cdots, \mathbf{x}_n \end{bmatrix} \in \mathbb{R}^{D \times n}$ represent the data sample matrix, where each column vector $\mathbf{x}_i$ ($i = 1, \cdots, n$) corresponds to a $D$-dimensional data sample, and $n$ denotes the total number of data samples. Moreover, let $\boldsymbol{\mu}$ represent the mean vector of all data samples, i.e., $\boldsymbol{\mu} = \frac{1}{n} \mathbf{X} \mathbf{1}_n$, where $\mathbf{1}_n$ is a column vector of ones of length $n$. For each class $j$ ($j = 1, \cdots, C$), let $\boldsymbol{\mu}^{(j)}$ represent the mean vector of the class, defined by $\boldsymbol{\mu}^{(j)} = \frac{1}{n^{(j)}} \mathbf{X}^{(j)} \mathbf{1}_{n^{(j)}}$, where $C$ is the total number of classes, $n^{(j)}$ is the number of samples in the $j$-th class, $\mathbf{X}^{(j)}$ is the submatrix of $\mathbf{X}$ containing only the samples of the $j$-th class, and $\mathbf{1}_{n^{(j)}}$ is a column vector of ones of length $n^{(j)}$.

Utilizing the above notations, the within-class scatter matrix, $\mathbf{S}_w$, which quantifies the aggregate variance within each class, and the between-class scatter matrix, $\mathbf{S}_b$, which measures the separation between different classes, can be defined as follows:

$$\mathbf{S}_w = \sum_{j=1}^{C} \sum_{\mathbf{x} \in \text{class } j} (\mathbf{x} - \boldsymbol{\mu}^{(j)})(\mathbf{x} - \boldsymbol{\mu}^{(j)})^T. \tag{25}$$

$$\mathbf{S}_b = \sum_{j=1}^{C} n^{(j)} (\boldsymbol{\mu}^{(j)} - \boldsymbol{\mu})(\boldsymbol{\mu}^{(j)} - \boldsymbol{\mu})^T. \tag{26}$$

LDA seeks to discover a projection matrix $\mathbf{W} \in \mathbb{R}^{D \times d}$ that transforms the original $D$-dimensional data into a lower $d$-dimensional subspace (where $d < D$), such that the projection, $\mathbf{W}^T \mathbf{X} \in \mathbb{R}^{d \times n}$, maximizes the separation between classes while minimizing the distance between samples within the same class. The objective function of LDA can be formulated as follows:

$$\max_{\mathbf{W}} \quad \frac{\text{tr}(\mathbf{W}^T \mathbf{S}_b \mathbf{W})}{\text{tr}(\mathbf{W}^T \mathbf{S}_w \mathbf{W})},$$
$$\text{s.t.} \quad \mathbf{W}^T \mathbf{W} = \mathbf{I}_d, \tag{27}$$

where $\mathbf{I}_d$ is the identity matrix of size $d$, and the constraint $\mathbf{W}^T \mathbf{W} = \mathbf{I}_d$ is introduced to ensure that the projection directions are orthonormal.

### A.3 Variants of Linear Discriminant Analysis

In this subsection, we review three typical categories of LDA variants: metric modification methods, max-min strategy-based methods, and neighborhood exploration methods. Each category represents a strategic adaptation aimed at enhancing the classic LDA method to address its inherent limitations.

Methods for metric modification aim to improve the LDA in terms of its between/within-class scatters, trace ratio objective, or distance measurement defined in the original objective function. Generally, metric modification methods seek to optimize the following:

$$\max_{\substack{\mathbf{W} = [\mathbf{w}_m]_{m=1}^d \in \mathbb{R}^{D \times d} \\ \mathbf{W}^T \mathbf{W} = \mathbf{I}}} \mathcal{P} \left( \frac{\mathcal{F}_{i,j} \left( \mathcal{G} \left( \mathbf{w}_m, \mathbf{S}_b^{i,j} \right) \right)}{\mathcal{H}_i \left( \mathcal{Q} \left( \mathbf{w}_m, \mathbf{S}_w^i \right) \right)} \right), \tag{28}$$

where $\mathbf{w}_m$ is the $m$-th column of the projection matrix $\mathbf{W}$, $\mathbf{S}_b^{i,j}$ is a specific between-class measurement for pushing the data samples from two distinct classes/subclasses $i$ and $j$ as far as possible, $\mathbf{S}_w^i$ is a within-class measurement for pulling the data points within the same class/subclass $i$ as close as possible, and $\mathcal{G}$, $\mathcal{Q}$, $\mathcal{F}_{i,j}$, $\mathcal{H}_i$, and $\mathcal{P}$ are some functions specifically designed in each algorithm. In conventional LDA, both $\mathcal{G}$ and $\mathcal{Q}$ are the standard quadratic functions with respect to $\mathbf{w}_m$. Meanwhile, $\mathcal{F}_{i,j}$ and $\mathcal{H}_i$ are the sum functions over their subscripts, respectively, and $\mathcal{P}$ is the identity function. Different from the traditional LDA, MGMD (Tao et al., 2008) defines $\mathbf{S}_b^{i,j}$ as the Kullback-Leibler (KL) divergence between the $i$-th and $j$-th classes. Additionally, $\mathcal{F}_{i,j}$ is defined as the geometric mean of the distances between all pairs of classes. For SDA (Zhu & Martinez, 2006), $\mathbf{S}_b^{i,j}$ is specified as the between-subclass scatter, in contrast to the traditional definition of between-class scatter used in the original LDA. In RSLDA (Wang et al., 2022a), the objective function is reformulated to improve the extraction of discriminant features. Here, $\mathcal{P}$ is defined as a summation function, while both $\mathcal{F}_{i,j}$ and $\mathcal{H}_i$ are treated as identity mappings. This formulation aims to compute the sum of the ratios of between-class scatter to within-class scatter across all individual dimensions of the latent subspace spanned by $\{\mathbf{w}_m\}_{m=1}^d$. In quadratic trace difference LDA (Wang et al., 2022b), $\mathcal{P}$ is defined as a quadratic function on the Stiefel manifold. In approaches that modify the distance measurement, the original LDA's use of Euclidean distance in $\mathbf{S}_b^{i,j}$ and $\mathbf{S}_w^i$ is substituted with alternative metrics. For instance, WDA (Flamary et al., 2018; Liu et al., 2020) employs the Wasserstein distance, also referred to as the optimal transport distance, while RLDA (Zhao et al., 2019) and $\ell_{2,1}$-LDA (Nie et al., 2021) utilize the $\ell_{2,1}$-norm distance metric.

Techniques based on the max-min strategy aim to separate different classes from the most challenging perspective (i.e., the closest classes) to alleviate the overlap caused by equally weighting all class pairs in traditional LDA. The formulation of this category of techniques can be represented as follows:

$$\max_{\mathbf{W}} \min_{i,j} \quad \mathcal{F} \left( \text{tr} \left( \mathbf{W}^T \mathbf{S}_b^{i,j} \mathbf{W} \right), \mathcal{G} \left( \text{tr} \left( \mathbf{W}^T \mathbf{S}_w' \mathbf{W} \right) \right) \right),$$
$$\text{s.t.} \quad \mathbf{W}^T \mathbf{W} = \mathbf{I} \tag{29}$$

where $\mathbf{S}_b^{i,j}$ denote the between-class scatter between classes $i$ and $j$, $\mathbf{S}_w'$ represents the within-class scatter, and $\mathcal{F}$ and $\mathcal{G}$ are defined differently in different methods. Specifically, both WLDA (Zhang & Yeung, 2010) and WDDR (Wang et al., 2024b) define $\mathbf{S}_b^{i,j}$ as the pairwise distance between class means, with $\mathcal{F}$ set as a ratio function. The difference lies in the definition of $\mathbf{S}_w'$ and $\mathcal{G}$. In WLDA, $\mathbf{S}_w'$ is specified as $\mathbf{S}_w^{(n)'}$ (where the subscript is not synchronized with $i$ or $j$) and represents the within-class scatter of a single class. Here, $\mathcal{G}$ is assigned as the maximum function among all candidates. In contrast, WDDR defines $\mathbf{S}_w$ as $\mathbf{S}_w^{(i,j)'}$, which denotes the pairwise joint within-class scatter, and $\mathcal{G}$ is simplified to an identity function. In HMMDA (Su et al., 2018), $\mathbf{S}_b^{i,j}$ is specified as the Chernoff distance scatter, which measures the entanglement of classes. Moreover, $\mathcal{G}$ is a sum function that aggregates the within-class scatter over all classes, similar to the design in conventional LDA.

Neighborhood exploration methods focus on modeling adjacent data points, so as to ensure the maximal preservation of the local geometric structure both within and across different classes. A general formulation for neighborhood exploration methods can be represented as follows:

$$\max_{\mathbf{W}} \quad \frac{\text{tr}(\mathbf{W}^T \widetilde{\mathbf{S}}_b \mathbf{W})}{\text{tr}(\mathbf{W}^T \widetilde{\mathbf{S}}_w \mathbf{W})},$$
$$\text{s.t.} \quad \mathbf{W}^T \mathbf{W} = \mathbf{I}. \tag{30}$$

Here, $\widetilde{\mathbf{S}}_b$ and $\widetilde{\mathbf{S}}_w$ represent the modified between-class and within-class scatter matrices, respectively, which are designed to characterize the relationships within neighborhoods. For example, in LFDA (Sugiyama, 2007), each data pair in $\widetilde{\mathbf{S}}_b$ and $\widetilde{\mathbf{S}}_w$ is weighted according to their affinity—the closer the two data points, the heavier the weight—unlike the equal weighting for all data pairs used in traditional LDA. The LLDA (Kim & Kittler, 2005) groups the dataset into $K$ local clusters, defines the $\widetilde{\mathbf{S}}_b$ and $\widetilde{\mathbf{S}}_w$ for each cluster, and performs the LDA within each cluster to obtain $K$ transformation matrices. New data samples are assigned to one of these $K$ clusters based on proximity to the cluster center, and the corresponding transformation matrix is used for dimensionality reduction. NMMP (Nie et al., 2007; Zhao et al., 2018) reformulates $\widetilde{\mathbf{S}}_b$ and $\widetilde{\mathbf{S}}_w$ by summing over pairs of nearby data samples, rather than over all data pairs as in the original LDA. Although LMNN (Weinberger & Saul, 2009) is not exactly equivalent to the trace-ratio formulation, it can still be interpreted in a similar local-scatter manner, where $\widetilde{\mathbf{S}}_w$ could be specified as the local within-class scatter over target-neighbor pairs, and $\widetilde{\mathbf{S}}_b$ could be specified as the local between-class scatter over active impostor pairs.

## B  Algorithm for NePuDA

---

**Algorithm 1** Optimization Procedure of NePuDA

---

**Input:** Training set $\{\mathbf{x}_i, \ell_i\}_{i=1}^n$
**Output:** Projection matrix $\mathbf{W}$
 1: Constructing $\mathcal{N}_i$ or $\mathcal{N}_i^+$ by Eq. (1) or Eq. (2);
 2: Constructing $\mathcal{N}_i^c$ by Eq. (3);
 3: Constructing $\mathbf{S}_i$ by Eq. (4);
 4: Constructing $\boldsymbol{\Sigma}$ by Eqs. (6-7);
 5: Randomly initialize $\mathbf{Z}$: $\mathbf{Z} \leftarrow \mathbf{Z}^{(0)}$, $t \leftarrow 1$;
 6: **while** not converged **do**
 7:     $\alpha_t \leftarrow \frac{\mathrm{tr}(\boldsymbol{\Sigma}\mathbf{Z}^{(t-1)})}{\max_i \mathrm{tr}(\mathbf{S}_i\mathbf{Z}^{(t-1)})}$;
 8:     Update $\mathbf{Z}^{(t)}$ by solving Eq. (15);
 9:     **if** $\left\|\mathbf{Z}^{(t)} - \mathbf{Z}^{(t-1)}\right\|_F \leq \varepsilon$ **then**
10:        $\mathbf{Z} \leftarrow \mathbf{Z}^{(t)}$, break;
11:     **else**
12:        $t \leftarrow t + 1$;
13:     **end if**
14: **end while**
15: Obtain $\mathbf{W}$ through the eigen-decomposition of $\mathbf{Z}$;
16: **return**  $\mathbf{W}$

---

## C  Connections to Existing Methods

In this section, we demonstrate the generalizability of the proposed method by analyzing its relationships to existing representative techniques in the field.

### C.1  Relationship with LDA (Fisher, 1936)

For the proposed method, if we remove the $\max_i$ operator from the denominator of the objective function in Eq. (8), set $k = n^{(j)} - 1$ for all data points within the $j$-th class ($j = 1, \cdots, C$), and replace $\boldsymbol{\Sigma}$ with the traditional between-class scatter $\mathbf{S}_b$, the proposed NePuDA will reduce to the traditional LDA.

### C.2  Relationship with SDA (Zhu & Martinez, 2006)

The proposed method shares a close relationship with the Subclass Discriminant Analysis (SDA). Specifically, if we remove the $\max_i$ operator from the denominator of our objective function, expand the neighborhood set to contain the entire dataset by setting $\mathbf{S}_i = \boldsymbol{\Sigma}_X$, where $\boldsymbol{\Sigma}_X$ is the data covariance matrix, and replace the between-class-neighborhood scatter $\boldsymbol{\Sigma}$ with the between-subclass scatter $\boldsymbol{\Sigma}_B$, i.e., substituting data samples with subclass means, then our method can be transformed into SDA.

### C.3  Relationship with WLDA (Zhang & Yeung, 2010)

As a representative of the max-min strategy category, the Worst-case LDA (WLDA) aims to separate the two closest classes to the best extent to reduce the overlap risk as much as possible by optimizing the following:

$$
\begin{aligned}
\max_{\mathbf{W}} \quad & \frac{\min_{i \neq j} \mathrm{tr}\left(\mathbf{W}^T\mathbf{S}_{ij}\mathbf{W}\right)}{\max_i \mathrm{tr}\left(\mathbf{W}^T\mathbf{S}_i\mathbf{W}\right)} \\
\text{s.t.} \quad & \mathbf{W}^T\mathbf{W} = \mathbf{I},
\end{aligned}
\tag{31}
$$

where $\mathbf{S}_{ij}$ denotes the pairwise scatter between the $i$-th class and the $j$-th class, and $\mathbf{S}_i$ denotes the within-class scatter of the $i$-th class. For our method, if we set the neighborhood size $k = n^{(j)} - 1$ for all data points

within the $j$-th class $(j = 1, \cdots, C)$, i.e., equally treating all data points from the same class, and replace the global between-class-neighborhood scatter $\mathbf{\Sigma}$ with the pairwise between-class scatter $\mathbf{S}_{ij}$, the objective function of the proposed method will be equivalent to that of the WLDA.

### C.4 Relationship with NMMP (Nie et al., 2007; Zhao et al., 2018)

Both Neighborhood MinMax Projections (NMMP) and our proposed method emphasize neighborhoods, but with a key distinction: NMMP treats all data points' neighborhoods equally, while our approach prioritizes neighborhoods of hard-to-classify points. Specifically, the objective function of NMMP can be expressed as follows:

$$\begin{aligned}
\max_{\mathbf{W}} \quad & \frac{\mathrm{tr}(\mathbf{W}^T \tilde{\mathbf{S}}_b \mathbf{W})}{\mathrm{tr}(\mathbf{W}^T \tilde{\mathbf{S}}_w \mathbf{W})} \\
\text{s.t.} \quad & \mathbf{W}^T \mathbf{W} = \mathbf{I},
\end{aligned} \tag{32}$$

where the between-class ($\tilde{\mathbf{S}}_b$) and within-class ($\tilde{\mathbf{S}}_w$) scatter matrices are defined as follows:

$$\tilde{\mathbf{S}}_b = \sum_{i,j:\mathbf{x}_i \in \mathcal{N}_j \,\&\, \mathbf{x}_j \in \mathcal{N}_i} (\mathbf{x}_i - \mathbf{x}_j)(\mathbf{x}_i - \mathbf{x}_j)^T,$$

$$\tilde{\mathbf{S}}_w = \sum_{i,j:\mathbf{x}_i \in \mathcal{N}_j^c \,\&\, \mathbf{x}_j \in \mathcal{N}_i^c} (\mathbf{x}_i - \mathbf{x}_j)(\mathbf{x}_i - \mathbf{x}_j)^T.$$

If we replace our within-class-neighborhood scatter $\mathbf{S}_i(i = 1, \cdots, n)$ and between-class-neighborhood scatter $\mathbf{\Sigma}$ by $\tilde{\mathbf{S}}_w$ and $\tilde{\mathbf{S}}_b$, respectively, and remove the $\max_i$ operator from the denominator of our objective function, i.e., equalizing the neighborhood purification across all samples, our method can be transformed into NMMP.

### C.5 Relationship with Recent Representation Learning

As we present in the paper, NePuDA can be used as a supervised feature extraction or representation refinement module in real applications, serving as a DR method which can boost image classifications (Wang et al., 2023; Yang et al., 2026), object identifications (Li et al., 2023), and integrate with deep models (Wang et al., 2024b). Recent advances in representation learning also exhibit conceptual connections to the proposed neighborhood-purification strategy. For example, Centerpolar (Yuan et al., 2026a) is a representative contrastive-learning and deep metric-learning method that aims to enhance local discriminability by polarizing class centers, thereby sharing similar intuition with the neighborhood-purification mechanism of NePuDA. Recent studies on noisy-label learning and uncertainty-aware representation refinement, such as CARE (Yuan et al., 2026b), further highlight the importance of handling ambiguous samples, which is also one of the main motivations of our proposed methods. In addition, the neighborhood-purification framework proposed in our work could be extended to several promising directions for handling more complex scenarios, such as spatio-temporal perception (Dang et al., 2025), video detection (Sheng et al., 2026), and direction-aware feature extraction (Jiang et al., 2023).

## D Complexity Analysis of SDP Optimization

In this section, we provide more details about the time complexity of SDP solver. For SDP solvers based on interior-point method (Bian & Tao, 2010), its time complexity is $\Theta\left(m_0^2 n_0^2\right)$, here $m_0 = D(D+1)/2 + 2$ is the number of variables, wherein $D(D+1)/2$ is the number of independent variables in the symmetric matrix $\mathbf{Z} \in \mathbb{R}^{D \times D}$, and 2 is for number of variables $s$ and $u$. Meanwhile, $n_0 = 2D + n + 4$ denotes the size of the problem, wherein $2D$ is for the two-side inequality $\mathbf{0} \preceq \mathbf{Z} \preceq \mathbf{I}_D$ in Eq. (15), $n$ is for the number of trace inequalities $\mathrm{tr}(\mathbf{S}_i \mathbf{Z}) \leq s, i \in [n]$, and 4 is for the trace equality $\mathrm{tr}(\mathbf{Z}) = d$ (which leads to inequalities $\mathrm{tr}(\mathbf{Z}) \leq d$ and $\mathrm{tr}(\mathbf{Z}) \geq d$), and the inequalities $\mathrm{tr}(\mathbf{\Sigma Z}) \geq u > 0$, in Eq. (15). Thus, supposing the iteration time is $\tau$, then the total SDP optimization complexity will be $\Theta\left(\tau m_0^2 n_0^2\right)$, where $m_0$ and $n_0$ are specified above.

# E   Description of Baseline Methods

This section specifically describes the selected baselines. We selected 14 representative dimensionality reduction methods for performance comparison, comprising 2 unsupervised methods and 12 supervised methods. The details of these methods are described below.

- Principal Component Analysis (PCA) (Hotelling, 1933): It is the most typical unsupervised dimensionality reduction method designed to preserve as much data information as possible by maximizing the data variance in the learned subspace.

- Feature-Sparse PCA (FSPCA) (Nie et al., 2023): It is a variant of PCA, which aims to find the sparse principal components by performing the variance maximization and feature selection simultaneously.

- Linear Discriminant Analysis (LDA) (Fisher, 1936): It is a classical supervised dimensionality reduction method that seeks to identify a subspace where the between-class distance is maximized, and the within-class variance is minimized.

- Fast Subclass Discriminant Analysis (FastSDA) (Chumachenko et al., 2021): This is a typical method within the metric modification category. It enhances efficiency and performance by dividing each class into multiple subclasses and utilizing the between-subclass Laplacian matrix for eigendecomposition.

- Ratio Sum LDA (RSLDA) (Wang et al., 2022a): This is a state-of-the-art method in the metric modification category. It improves class separability in the learned subspace by maximizing the sum of the ratios of between-class scatter to within-class scatter across each dimension.

- Trace Ratio LDA (TRLDA) (Wang et al., 2022b): This is another method within the metric modification category. It reformulates the trace ratio objective function in the original LDA into a quadratic optimization problem on the Stiefel manifold.

- Robust LDA (RLDA) (Zhao et al., 2019): This method also falls within the metric modification category. Instead of the traditional $\ell_2$-distance, RLDA employs the $\ell_{2,1}$-distance in the calculation of within-class scatters. This adjustment has been demonstrated to enhance robustness against outliers.

- $\ell_{2,1}$-LDA (Nie et al., 2021): This method is an advancement of RLDA. It aims to improve discriminability by not only minimizing the within-class scatter but also maximizing the total sample scatter.

- Worst-case LDA (WLDA) (Zhang & Yeung, 2010): This method employs a max-min strategy. Instead of using average distances, WLDA maximizes the minimum between-class scatter and minimizes the maximum within-class scatter. This ensures that the distance between each class pair is maximized, while each pairwise distance within the same class is minimized.

- Heteroscedastic Max–Min Distance Analysis (HMMDA) (Su et al., 2018): This method also utilizes the max-min strategy. It maximizes the pairwise Chernoff distance between the closest classes while minimizing the within-class scatter.

- Worst-case Discriminative Dimensionality Reduction (WDDR) (Wang et al., 2024b): This is a cutting-edge approach that leverages the max-min strategy. This method aims to maximize the trace ratio of the two least separable classes, thereby enhancing the discriminative power of the reduced-dimensional representation.

- Neighborhood Component Analysis (NCA) (Goldberger et al., 2004): It's a supervised neighborhood-based metric learning method that learns a linear transformation by maximizing the expected leave-one-out nearest-neighbor classification performance.

- Large Margin Nearest Neighbor (LMNN) (Weinberger & Saul, 2009): It's a method categorized into neighborhood exploration that learns a Mahalanobis distance by pulling target neighbors together and pushing differently labeled impostors away with a large margin.

- Adaptive Neighborhood MinMax Projections (ANMMP) (Zhao et al., 2018): It is a method that falls within the category of neighborhood exploration techniques. Instead of analyzing entire classes in a coarse manner, ANMMP operates at a finer scale, focusing on the local structure of the dataset. Its primary goal is to bring homogeneous nearby data samples closer together while distancing points from different classes.

## F  Parameter Sensitivity Analysis of NePuDA

This section examines the sensitivity of two versions of our proposed methods, NePuDA and NePuDA$^+$, with respect to two key parameters: the neighborhood size ($k$) and the reduced dimension ($r.d.$), which shows the robustness across different scenarios. Specifically, we evaluate the classification accuracy on three datasets, Segment, Libras, and YaleB, across various parameter combinations. The neighborhood size $k$ is tested from 2 to 16 (except for NePuDA on YaleB, since it requires larger value of $k$, thereby needs broader range of exploration, from 6 to 48), while $r.d.$ is explored across a range containing the optimal performance-yielding dimension. The results are presented in Fig. 6. As we can see, both NePuDA and NePuDA$^+$ demonstrate robust performance across all three datasets under various combinations of $k$ and $r.d.$. This observation is consistent with the findings from our previous synthetic experiment, showing the reliability and adaptability of our proposed approaches in diverse data environments. Meanwhile, as we observe from YaleB which contains larger amount of data points, NePuDA$^+$ performs better for smaller $k$ (e.g., for $k = 6$, the mean accuracy of NePuDA across reduced dimensions is 75.60%, while for NePuDA$^+$, it's 85.39%), while NePuDA performs more competitive with larger $k$. Such observation aligns well with the rationale of parameter settings that introduced in Section 5.3.2.

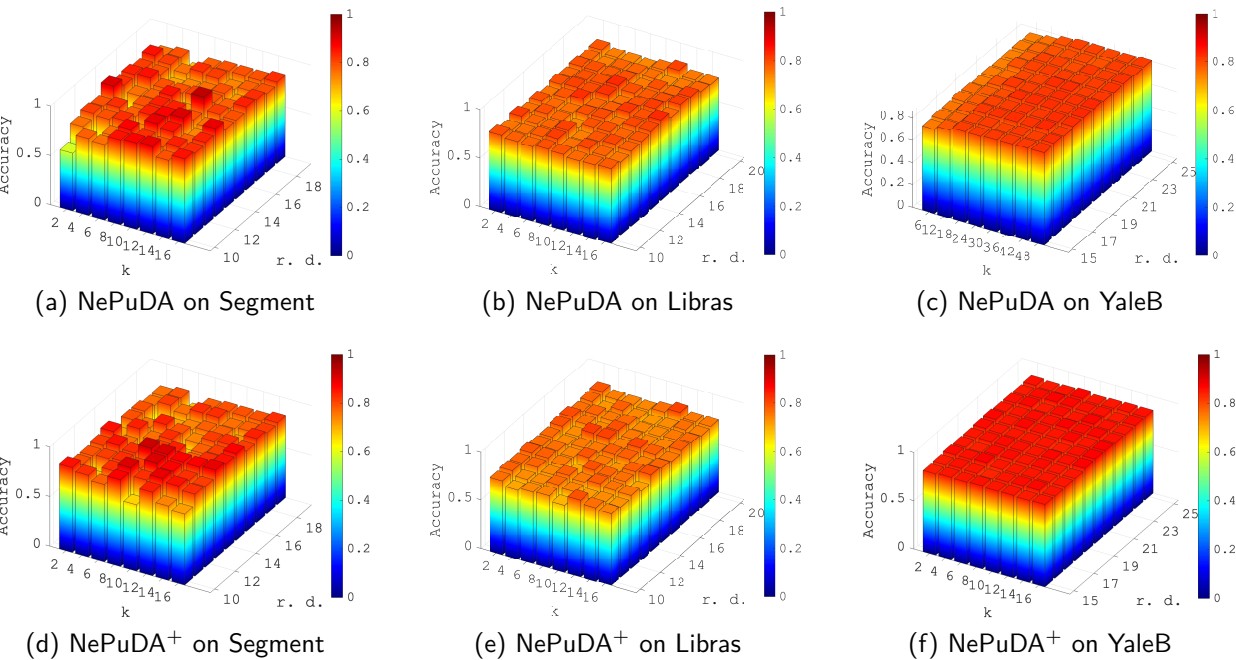

(a) NePuDA on Segment  (b) NePuDA on Libras  (c) NePuDA on YaleB

(d) NePuDA$^+$ on Segment  (e) NePuDA$^+$ on Libras  (f) NePuDA$^+$ on YaleB

Figure 6: Classification accuracy of NePuDA and NePuDA$^+$ on Segment, Libras, and YaleB with various combinations of neighborhood size ($k$) and reduced dimension ($r.d.$).

## G  Convergence Analysis of NePuDA

In this section, we conduct the convergence analysis of the proposed method. For the optimization process, we employ the MATLAB CVX package to solve the SDP problem formulated in Eq. (15) at each iteration. Specifically, we utilize SDPT3 as the solver within the CVX framework (Grant et al., 2009). We demonstrate the results on four datasets, Hayes-roth, Glass, Segment, and Libras, in Fig. 7. The y-axis of the figure is the absolute value of CVX variable 'cvx_optval'. It gives the absolute value of the objective function after CVX completes at each iteration. As can be seen in the figures, the curves in all scenarios eventually approach zero. This shows that the absolute difference in the objective function between two consecutive iterations, i.e., $|h(\mathbf{Z}^{(k)}) - h(\mathbf{Z}^{(k-1)})|$, tends to zero as $k$ increases. Such behavior provides clear empirical evidence of the convergence of the algorithm. The results demonstrate the efficiency of both versions of our proposed method. Across all datasets examined, the algorithms consistently converge in around 10 iterations, showing the practical applicability of the proposed method to a wide range of datasets.

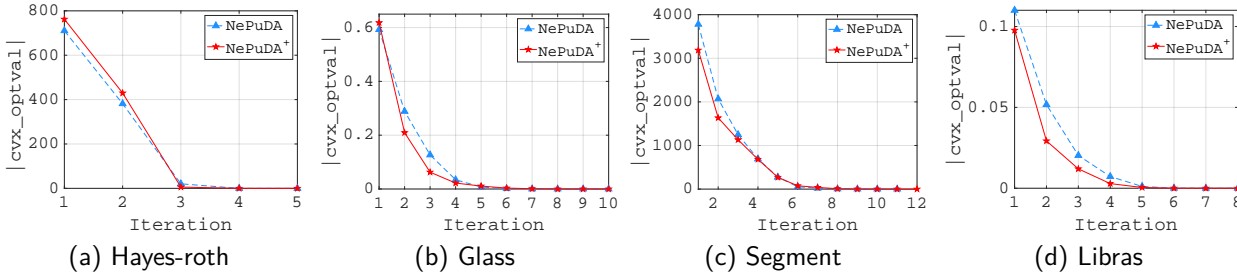

| (a) Hayes-roth | (b) Glass | (c) Segment | (d) Libras |

Figure 7: The convergence curve of two versions of the proposed method, NePuDA and NePuDA$^{+}$, on four datasets: Hayes-roth, Glass, Segment, and Libras.

## H  Key Statistics of Selected Datasets

In this section, we briefly summarize the statistics information of selected datasets, which is presented below in Table. 4. (Datasets with $^{*}$ are adopted in Appendix I.)

Table 4: Statistics of 13 datasets used in our experiments.

| Dataset | Size ($n$) | Dimensionality ($D$) | Class Number ($C$) |
|---|---|---|---|
| Iris | 150 | 4 | 3 |
| Spiral | 312 | 2 | 3 |
| Hayes-roth | 132 | 5 | 3 |
| Yeast | 1484 | 8 | 10 |
| Glass | 214 | 9 | 6 |
| Balance-scale | 625 | 4 | 3 |
| Sonar | 208 | 60 | 2 |
| Segment | 210 | 19 | 7 |
| Libras | 360 | 90 | 15 |
| Ionosphere | 351 | 34 | 2 |
| JAFFE | 213 | 256×256 | 7 |
| Yale | 165 | 32×32 | 15 |
| YaleB | 2414 | 32×32 | 38 |
| Isolet$^{*}$ | 7797 | 617 | 26 |
| MNIST$^{*}$ | 70000 | 28×28×1 | 10 |
| FashionMNIST$^{*}$ | 70000 | 28×28×1 | 10 |

# I  Validations on Large-Scale Datasets with Various Downstream Evaluations and Explorations on Integration with Deep Pre-Trained Models

In this section, we adopt three relatively large-scale datasets to further demonstrate the capability of the proposed methods in handling large-scale data. Specifically, Isolet dataset is a high-dimensional speech dataset consisting of isolated spoken English letters (A–Z) from multiple speakers, with 7,797 samples and 26 classes. MNIST contains handwritten digits, with clear visual patterns and background. FashionMNIST consists of images of clothing items from different categories, which involves richer shape variations and more detailed visual structures. Both MNIST and FashionMNIST contain 70,000 samples with 10 classes.

Meanwhile, to explore the integration of the proposed methods with deep learning models, we adopt a deep pre-trained model, ResNet34 (He et al., 2016) model, with a fully connected linear layer ($512 \times 128$) in the end to obtain a 128-dimensional embedding, for MNIST and FashionMNIST datasets. Specifically, for each dataset, we freeze the pre-trained ResNet34 model, and add a $512 \times 128$ fully connected layer and a classification head in the end. Then we train the model on the dataset, with early stopping strategy. As the early stopping is triggered, we terminate the training process and remove the classification head, using the trained model to obtain the 128-dimensional deep embedding of each data sample. Afterwards, we use PCA as a filter to remove noises. Specifically, for MNIST dataset, PCA reduces 128-$d$ embeddings to 40-$d$ features which preserves 97.04% of the total energy, while reduces 128-$d$ embeddings to 50-$d$ features which preserves 97.75% of the total energy for FashionMNIST. Following this, we use different DR methods to obtain the projection of each sample, and use downstream classifiers to obtain the average classification accuracy across 5 trials and the corresponding standard deviation. To be comprehensive, here we use more kinds of classifiers to avoid being uncommon for large-scale classification. Specifically, the classifiers adopted here are: 3-NN (following the main paper), support vector machine (SVM), and random forest (RF). In addition to 3-NN, SVM focuses on learning decision boundaries between classes, which naturally aligns with the notion of neighborhood purification. Specifically, well-purified neighborhoods lead to clearer and more separable interfaces, which are easier for SVM to capture. RF serves as a general classifier, which could capture data patterns at multiple levels of granularity. Notably, both classifiers are classical and appropriate in this task which avoid overwriting the discriminativeness of the solved subspace.

For all new datasets, we select 6 baselines from the summarized categories, and they are LDA, FastSDA, TRLDA, WLDA, WDDR, and ANMMP. Since the experiments on Isolet follows the tradition which are not processed by pre-trained models, we still select follows the original setting (i.e., we select 70% of the total samples for training and left for testing). For MNIST and FashionMNIST, we select 20% of the total samples for training and left for testing. For downstream classifiers, we use 3-NN, SVM, and RF. For SVM, we adopt radial basis function (RBF) kernel, where the kernel scale is set to 10. For RF, we use an ensemble of 50 decision trees constructed via bootstrap aggregation (bagging). Each tree is trained on a random subset of the data, and the final prediction is obtained by majority voting among all trees. For better presentation of performance, we use bars with different colors to denote the mean accuracy of different methods, while the error bar denotes the corresponding standard deviation.

As observed in Fig. 8, for each method, different classifiers exhibit comparable performance on the learned projections of different methods. For comparison, NePuDA$^+$ mostly outperforms the selected baselines in terms of mean accuracy across all classifiers, and NePuDA achieves a competitive performance on MNIST and FashionMNIST dataset, while demonstrating the potential to capture discriminative features on large-scale datasets.

In addition, we further explore the role of the pre-trained deep model, by comparing the results of settings with pre-trained deep model against that of original settings in the main paper. To this matter, for MNIST and FashionMNIST datasets, we conduct new experiments that take our traditional settings as a comparison. Following the previous settings, independent from deep models, we directly use PCA as a filter to preserve at least 90% energy (90-$d$ for MNIST with 90.42% energy preserved, and 85-$d$ for FashionMNIST with 90.14% energy preserved), and we adopt the dimensionality reduction methods on the principal components, finally use three classifiers to obtain the mean classification accuracy across 5 trials and the corresponding standard deviation. We also use 20% of the total samples for training, and the left for testing, and this is for fair comparison with the previous experiments on the two datasets. Fig. 9 presents the comparison results, where

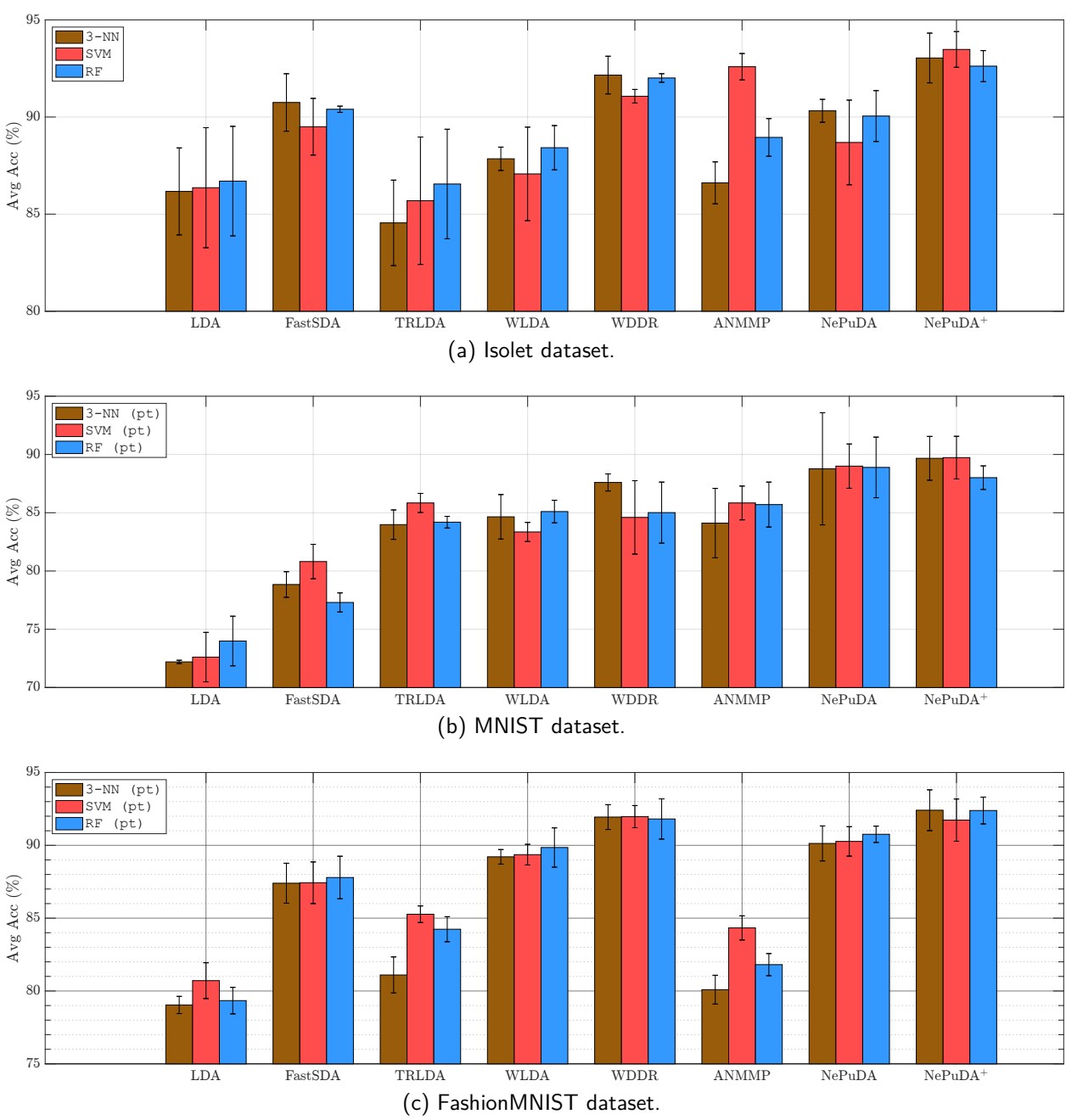

Figure 8: The mean accuracy bar of different methods with the error bar denotes the standard deviation on three large datasets.

**pt** denotes the strategy with pre-trained models, and **trad** denotes the traditional strategy, with the same color bar as the corresponding classifier but with hatching.

As shown in Fig. 9a, on FashionMNIST, the pretrained-model-based training strategy consistently achieves higher classification accuracy than the traditional training strategy for every method. By contrast, the opposite trend is observed on MNIST, as shown in Fig. 9b. The possible reason is that FashionMNIST contains more complex visual patterns that require a higher-level understanding of shape and texture, which deep pre-trained models can effectively capture, thereby improving classification accuracy. On the other hand, MNIST has a clear background and simple image composition, and the use of a pre-trained model

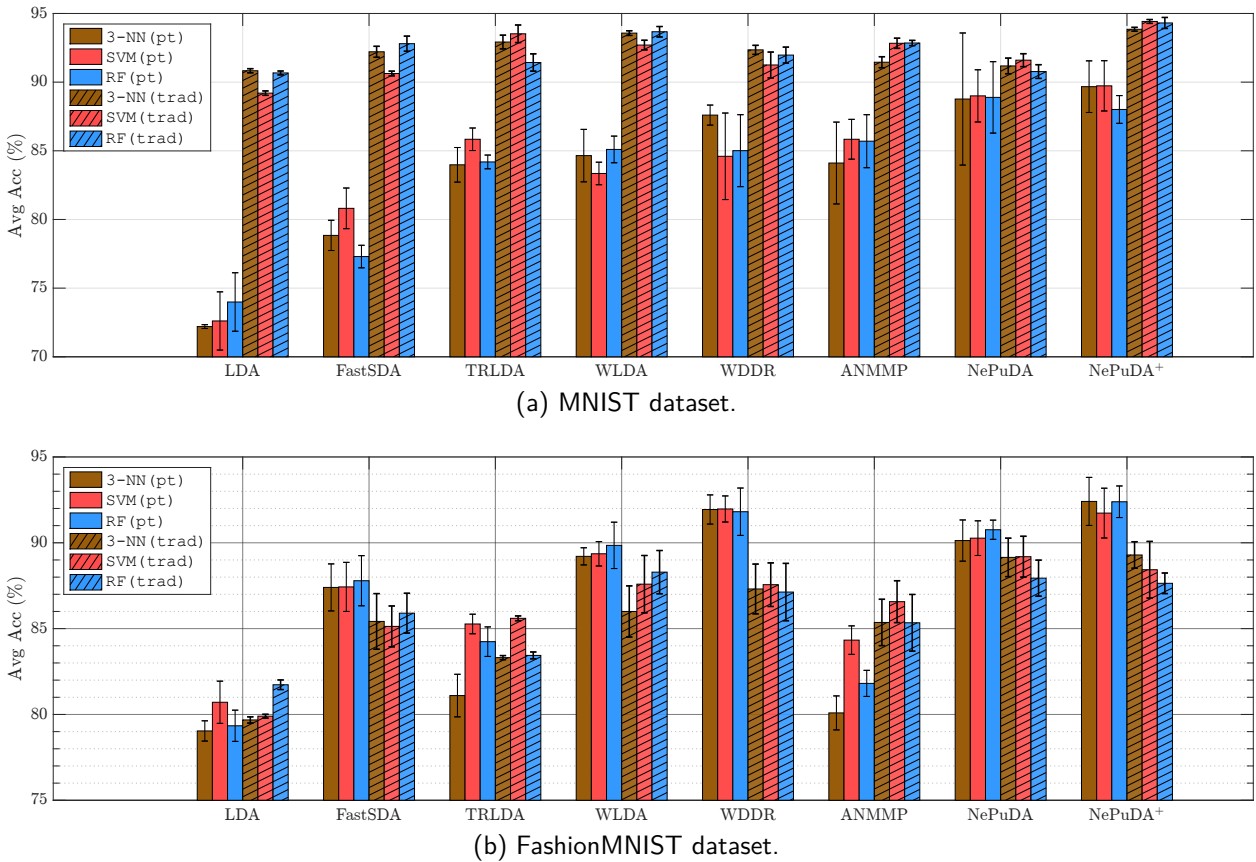

Figure 9: The mean accuracy bar of different training strategies with the error bar denotes the standard deviation on the two large datasets.

may provide more limited benefits, whereas PCA preserves the global variance structure without introducing additional semantic bias, making it more suitable for subsequent dimensionality reduction in such scenarios. This is an interesting observation, as it suggests that deep models can enhance DR methods in capturing complex data characteristics.

## J  Visualization of Samples from JAFFE Dataset

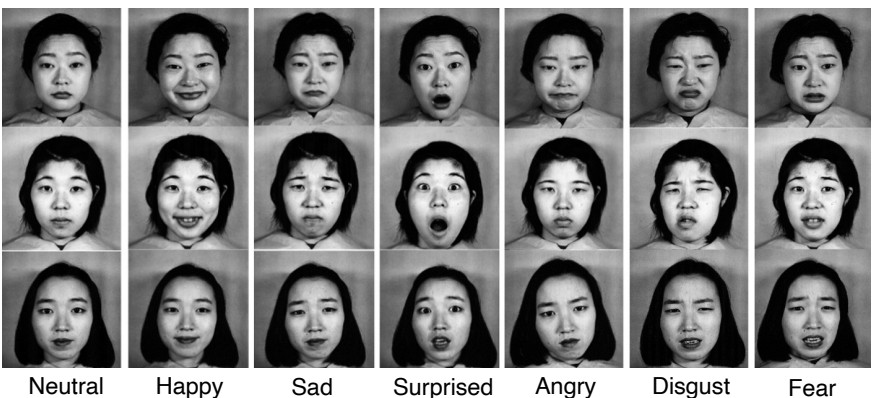

Neutral  Happy  Sad  Surprised  Angry  Disgust  Fear

Figure 10: Samples from the JAFFE dataset.

## K  Running Time of Proposed Methods

In this section, we summarize the running time of our two versions of proposed methods, which could provide much stronger evidence for the method's practical applicability. Specifically, we select YaleB dataset to conduct our experiments, and we investigate the empirical upper bound of runtime with respect to both the number of training samples and the input dimensionality. To this end, we consider two sample sizes, i.e., 1000 and 2000. For each setting, the input dimensionality is traversed in {50, 75, 100, 125, 150, 200}, which is implemented by preserving certain number of principal components. We record the running time of both algorithms that they finish one trial. All experiments are conducted by MATLAB R2022b on a Linux server (Oracle Linux Server 8.10) equipped with 256 GB RAM. The runtime results of both methods are reported in Table 5. As observed, both algorithms are efficient when the training data is with tens of dimensionality. As the input dimensionality increases to several hundred, the runtimes of both methods grow substantially, reaching approximately 20 hours or more. This can be regarded as the practical upper limit of the current implementations. Moreover, the difference in runtime between the two methods primarily arises

Table 5: Running time of both versions of proposed methods on YaleB dataset with different numbers of training data and various input dimensionality.

| #Num. of Train. Data | @Input Dim. | NePuDA | NePuDA$^+$ |
|---|---|---|---|
| #1k | @50 | 2m 17.24s | 3m 31.79s |
| | @75 | 9m 14.55s | 15m 37.96s |
| | @100 | 27m 9.69s | 43m 30.27s |
| | @125 | 1hr 12m 6.19s | 2hrs 50m 7.91s |
| | @150 | 2hrs 40m 9.79s | 5hrs 27m 10.61s |
| | @200 | 8hrs 37m 36.12s | 14hrs 28m 1.01s |
| #2k | @50 | 6m 15.15s | 9m 23.99s |
| | @75 | 24m 23.91s | 38m 12.64s |
| | @100 | 1hr 0m 22.49s | 1hr 47m 52.20s |
| | @125 | 3hrs 50m 2.53s | 6hrs 31m 20.62s |
| | @150 | 6hrs 18m 9.77s | 13hrs 48m 15.99s |
| | @200 | 17hrs 22m 21.76s | 31hrs 7m 7.03s |

from their different strategies for constructing the within-class-neighborhood scatter matrices (see Section 4.2 for details).

To improve scalability to large-sample and high-dimensional data, future work may explore approximate incremental optimization strategies. One possibility is to optimize the columns of $\mathbf{W}$ sequentially, reducing each subproblem from matrix to vector optimization. This enables the use of efficient DC (difference-of-convex) optimization frameworks, such as constrained CCCP (constrained concave-convex procedure). Although such methods improve computational efficiency, they may converge to stationary points and thus obtain suboptimal solutions. Therefore, they are more suitable for large-scale settings, whereas the current strategy remains preferable when sample size is moderate with higher accuracy requirement.

