# OpenReview forum: "NePuDA: Neighborhood-Purifying Discriminant Analysis"
_TMLR — Under review for TMLR_

### Review · Reviewer_8jLu · 2026-04-05

**Summary Of Contributions:**

The paper introduces Nepuda, a linear supervised dimensionality reduction method that tries to improve upon Linear Discriminant Analysis (LDA). The idea is the method is to find a linear subspace where classes are locally separated (rather than globally), which in the paper is described by saying that each neighborhood of a point is pure (i.e., the k-nearest neighbors should have the same label). Two variants of the methods are defined, with different neighborhood constrcutions. The optimization problem is solved by repeatedly solving semi-definite programming (SDP) problems, and convergence guarantees are provided. The method is benchmarked on some small datasets, including synthetic and real-world ones, and shows competitive results.

Strengths:
- the idea of seeking local separation rather than global one is really interesting
- the empirical results, albeit preliminary, show competitive results on both synthetic and real-world data
- the theoretical analysis is sound

Weaknesses:
- running SDP several times is really expensive, it might be limiting on large datasets
- it is unclear how reasonable values of $k$ should be selected. Is using a validation dataset the best choice? How did you select the values of $k$ in the experiments? A discussion on this should be added.
- using a 3-NN classifier for classification is very unrealistic in practice. This is a major limitation for the relevance of the paper.
- similarly to the point above, the real-world datasets are small and outdated. I am not an expert in the field, but I think that using the method on pre-trained embeddings from large models for classification (e.g. like for linear probing) would be interesting.
- the gains in the classification setting are marginal, and no nonlinear methods are benchmarked (although I see that this might be out of scope)

**Audience:**

Yes

**Audience Explanation:**

The method is an interesting improvement over LDA.

**Claims And Evidence:**

Yes

**Claims Explanation:**

The idea is interesting, the technical analysis and computational analysis is sound. The experimental evaluation is limited, but seemingly sound.

**Requested Changes:**

- add a discussion on the selection of $k$
- if possible, add some more relevant or larger datasets
- similarly, add if possible some more practical classifiers

---

> ### Author Response · Authors · 2026-06-16
> **Response to Reviewer 8jLu (1)**
>
> We are sincerely grateful for the reviewer’s commitment of valuable time in carefully evaluating our manuscript and for recognizing our idea, empirical results, and the theoretical analysis. We also greatly appreciate the constructive feedback and detailed suggestions expertly provided on the parameter settings and experimental designs. Below, we respond to the reviewer’s valuable feedbacks. All revisions made in the manuscript are highlighted in blue for easy identification.
>
> ---
>
> * Regarding the computational cost of SDP and its applicability to large datasets, we appreciate the reviewer's comment. Yes, we agree with the reviewer that running SDP is expensive. The reason for designing the optimization strategy based on SDP optimization is that it ensures a globally optimal solution at each iteration of our algorithm and converges to the global optimum of the relaxed problem, which is important for achieving reliable neighborhood purification.
>
>
>   In our future work, we plan to develop more efficient alternatives to the proposed method to enhance its scalability for large-scale datasets. For example, column optimization based on constrained concave convex optimization (CCCP) and second-order cone programming (SOCP) are more efficient for large-scale cases, since the optimization variables are reduced to vectors rather than matrices, which implies that we optimize each column of the whole matrix rather than the whole matrix. By doing so, we can reduce computational complexity compared to SDP while preserving key properties of the model. In addition, we are also interested in developing surrogate loss functions that approximate the proposed objective and can be integrated into deep learning architectures. This would allow us to leverage GPU acceleration and extend the method to large-scale settings more efficiently.
>
> ---
>
> * Regarding the parameter of the neighborhood set (i.e., $k$), we appreciate the reviewer's insightful suggestions for clarifying its setting. In our experiments, following standard machine learning practice for parameter selection, we use a validation set to determine the neighborhood size $k$ for each experiment.
>   Here, the parameter $k$ is empirically selected by searching over a predefined range, as described in Section 5.3.2 (Experimental Setup) in the manuscript. Specifically, $k$ is chosen as a multiple of $\lfloor \min_{j=1,\dots,C} n^{(j)} / 2 \rfloor$ (i.e., $\mathrm{const}\times \lfloor \min_{j=1,\dots,C} n^{(j)} / 2 \rfloor$), where $n^{(j)}$ denotes the sample size of the $j$-th class. In terms of the range of the parameter $k$, since NePuDA first selects neighboring data points and then picks same-class data points, a relatively larger $k$ (compared to that of NePuDA$^+$) is needed to ensure sufficient within-class neighbors are considered, and $\\mathrm{const}\\in\\left \\{ 1,2,3,4 \right \\} $. For NePuDA$^+$, it starts from same-class data points and then considers neighboring data points, so a smaller $k$ is generally sufficient to build up the within-class-neighborhood scatter. Therefore, $\\mathrm{const}\\in\\left \\{ 1,\\frac{1}{2} ,\\frac{1}{2^2},\\dots,\\frac{1}{2^{i-1}}  \\right \\}  $ where $i \le \log_2\min_{j=1,\dots,C}^{}n^{(j)}-1$ to make $k$ is no smaller than 2. Based on the searching range, in our implementation, the optimal value of neighborhood size $k$ is determined based on the mean accuracy of 5-fold cross validation, to avoid any potential bias.
>
>   **Revisions:** We describe the selection range of $k$, together with its parameter settings in highlighted part of Section 5.3.2 (Experimental Setup), from line 410 to line 414.

---

> ### Author Response · Authors · 2026-06-16
> **Response to Reviewer 8jLu (2)**
>
> * Regarding the setting about using 3-NN as the downstream classifier, we thank the reviewer for this insightful comment. The reason that we adopt $k$NN classifier for downstream evaluation is that we follow a common standard in dimensionality reduction community [1,2,3], with $k=3$ referring to existing literatures [4,5]. Despite this, we agree with the reviewer that $k$NN classifier is relatively simple. The reason that we employ such a simple classifier (rather than more powerful alternatives) is to demonstrate that the strong classification performance is attributable solely to the discriminativeness of the discovered subspace, rather than to the inherent capabilities of a sophisticated classifier.
>
>   We agree that a more comprehensive evaluation with various classifiers would provide more convincing evidence of the effectiveness of our proposed methods. Therefore, we conduct further experiments with different downstream classifiers. Specifically, we evaluate (1) the classification performance of the $k$NN classifier with different values of $k$; (2) the classification performance across different types of classifiers.
>
>   (1) We perform a comparison using the $k$NN classifier with varying values of $k$ ($k\\in\\left\\{3,5,7\\right\\}$) on 3 datasets: Iris, Glass, and Libras, with increasing numbers of data points and original dimensionality. We add the results for $k = 5$ and $k = 7$ to the previously reported results. The experimental settings are identical to those in the original manuscript. For comparison, we select 3 baseline methods, one from each of the 3 categories summarized in the manuscript. The results are presented below.
>
>   ````markdown
>   ---------------------------------------------------------------------------
>   Method    Classifier  Metric     Iris          Glass         Libras
>   ---------------------------------------------------------------------------
>   FastSDA   3-NN        Accuracy   96.00±1.86    68.92±4.13    80.00±1.90
>                         r.d.       d=2           d=4           d=12
>             5-NN        Accuracy   98.22±1.99    68.62±3.54    78.15±5.95
>                         r.d.       d=2           d=5           d=10
>             7-NN        Accuracy   96.89±1.99    68.31±5.40    77.78±6.34
>                         r.d.       d=2           d=5           d=10
>   ---------------------------------------------------------------------------
>   WLDA      3-NN        Accuracy   96.00±0.99    69.54±5.48    76.38±5.19
>                         r.d.       d=3           d=7           d=13
>             5-NN        Accuracy   96.89±1.22    65.54±2.33    72.96±1.98
>                         r.d.       d=2           d=8           d=14
>             7-NN        Accuracy   96.44±2.53    63.69±6.21    69.81±3.85
>                         r.d.       d=2           d=7           d=14
>   ---------------------------------------------------------------------------
>   ANMMP     3-NN        Accuracy   96.00±2.90    69.85±4.69    80.38±3.73
>                         r.d.       d=2           d=8           d=12
>             5-NN        Accuracy   97.33±2.43    70.46±4.67    78.33±4.40
>                         r.d.       d=2           d=7           d=13
>             7-NN        Accuracy   96.89±2.53    68.31±6.12    82.04±4.74
>                         r.d.       d=1           d=6           d=17
>   ---------------------------------------------------------------------------
>   NePuDA    3-NN        Accuracy   97.04±1.28    66.67±7.11    80.19±3.12
>                         r.d.       d=1           d=8           d=14
>             5-NN        Accuracy   96.44±2.53    66.15±8.45    82.41±4.04
>                         r.d.       d=2           d=8           d=17
>             7-NN        Accuracy   96.44±2.53    69.23±6.21    79.44±1.35
>                         r.d.       d=2           d=8           d=18
>   ---------------------------------------------------------------------------
>   NePuDA+   3-NN        Accuracy   98.52±1.28    70.46±4.13    81.43±2.52
>                         r.d.       d=3           d=8           d=15
>             5-NN        Accuracy   96.89±1.99    76.92±3.00    80.56±3.21
>                         r.d.       d=3           d=6           d=16
>             7-NN        Accuracy   97.33±1.86    70.77±6.21    82.41±1.65
>                         r.d.       d=3           d=8           d=17
>   ---------------------------------------------------------------------------
>   ````
>
>   We observe that the classification performance on the subspace projections is robust to different choices of $k$ in the $k$NN classifier. Moreover, our proposed methods consistently outperform the baselines in most cases across the tested values of $k$, which demonstrates the effectiveness of our methods.

---

> ### Author Response · Authors · 2026-06-16
> **Response to Reviewer 8jLu (3)**
>
> (2) We further evaluate the robustness of our subspace projections using two additional downstream classifiers: support vector machine (SVM) and random forest (RF). SVM is designed to learn optimal decision boundaries between classes, which naturally complements the concept of neighborhood purification. Specifically, well-purified neighborhoods produce clearer and more separable class interfaces, making them easier for SVM to capture effectively. In contrast, RF serves as a versatile ensemble classifier capable of capturing complex data patterns at multiple levels of granularity. As observed, our proposed methods, especially NePuDA$^+$, outperform the baselines in most cases across different classifiers, further validating the effectiveness of our proposed methods.
>
> **Revisions:** We present additional experimental results on relatively large datasets using SVM and Random Forest classifiers, in addition to the $k$NN classifier, in Appendix I.
>
> ---
>
> * Regarding the datasets, we sincerely appreciate the reviewer's suggestion, and we totally agree with the reviewer that validating our proposed methods on pre-trained embeddings from large models would be interesting. To take the reviewer's advice into consideration, we perform additional experiments using pre-trained deep features extracted from relatively large datasets via deep neural networks. Specifically, our new datasets include 1 traditional large-scale dataset, Isolet (with 7797 samples), and 2 image datasets, MNIST and FashionMNIST. MNIST contains handwritten digits, with clear visual patterns and background. FashionMNIST consists of images of clothing items from different categories, which involve richer shape variations and more detailed visual structures. The statistics of the new datasets are presented in Table 4 of the manuscript.
>
>
>   For MNIST and FashionMNIST, we use the pre-trained ResNet34 [6] model, with a fully connected linear layer (512$\times$128) in the end to obtain a 128-dimensional embedding. Afterward, we use different DR methods to obtain projections (with 20\% of the total samples for training and the remainder for testing) and use 3 classifiers (i.e., 3-NN, SVM with an RBF kernel, and RF with 50 bagged trees) to perform the classification task. For Isolet, we follow the settings in the manuscript, and we select 70\% of the total samples for training and the remainder for testing, also evaluated with 3 classifiers. For all new datasets, we select 6 baselines from the summarized categories, and they are LDA, FastSDA, TRLDA, WLDA, WDDR, and ANMMP.
>
>   As observed in Figure 8 in the manuscript, for each method, different classifiers exhibit comparable performance on the learned projections. For comparison, NePuDA$^+$ mostly outperforms the selected baselines in terms of mean accuracy across all classifiers, and NePuDA achieves a competitive performance on MNIST and FashionMNIST dataset, demonstrating the potential to capture discriminative features on large-scale datasets.
>
>   In addition, we further explore the role of the pre-trained deep model, by comparing the results of settings with pre-trained deep model against that of original settings in our manuscript. To this matter, for MNIST and FashionMNIST datasets, we conduct new experiments based on our traditional settings as a comparison. The Figure 9 in the manuscript presents the comparison results, where $\mathtt{pt}$ denotes the strategy with pre-trained models, and $\mathtt{trad}$ denotes the traditional strategy, with the same color bar as the corresponding classifier but with hatching. Interestingly, as we observed in Figure 9(a) in the manuscript, the pre-trained-model-based training strategy consistently achieves higher classification accuracy than the traditional training strategy on FashionMNIST, but lower accuracy on MNIST. This is mainly because FashionMNIST contains more complex visual patterns that require a higher-level understanding of shape and texture, which deep pre-trained models can effectively capture, thereby improving classification accuracy. By contrast, MNIST has a clear background and simple image composition, and the use of a pre-trained model may provide more limited benefits. This is an interesting observation, as it suggests that deep models can enhance DR methods in capturing complex data characteristics, which aligns well with the reviewer's suggestion.
>
>   **Revisions:** We perform additional experiments on large datasets with different classifiers in Appendix I, to adapt our methods to deep architectures, which improves the applicability of our proposed methods.

---

> ### Author Response · Authors · 2026-06-16
> **Response to Reviewer 8jLu (4)**
>
> * Regarding the performance gain in the classification tasks, we acknowledge that the performance improvements achieved by our method are consistent but modest. Currently, our primary objective is to demonstrate the potential of the proposed methods in extracting discriminative information from complex and irregular data distributions within a linear setting.
>
>   We appreciate the reviewer for the comment on benchmarking our methods against nonlinear baselines. Based on the reviewer’s valuable suggestions, we adapt our methods to nonlinear settings by applying them to deep embeddings and evaluating the resulting representations with nonlinear downstream classifiers. The experimental results provide preliminary evidence that our methods can be effectively extended to nonlinear scenarios. Additionally, in future work, we plan to nonlinearize our methods through kernelization or by designing appropriate loss functions to adapt them to deep learning architectures, in order to further improve its effectiveness for classification tasks. We also intend to compare our nonlinearized method with non-linear dimensionality reduction methods, as well as deep learning approaches, to further demonstrate its effectiveness.
>
> **References:**
>
> [1] W. Bian and D. Tao. "Max-min distance analysis by using sequential SDP relaxation for dimension reduction". In: IEEE Trans. Pattern Anal. Mach. Intell. 33.5 (2010), pp. 1037–1050.
>
> [2] Y. Zhang and D.-Y. Yeung. "Worst-case linear discriminant analysis". In: Adv. Neural Inf. Process. Syst. 23 (2010), pp. 2568-2576.
>
> [3] Z. Wang et al. "Worst-Case Discriminative Feature Learning via Max-Min Ratio Analysis". In: IEEE Trans. Pattern Anal. Mach. Intell. 46.1 (2024), pp. 641–658.
>
> [4] J. Wang et al. "Ratio Sum Versus Sum Ratio for Linear Discriminant Analysis". In:
> IEEE Trans. Pattern Anal. Mach. Intell. 44.12 (2022), pp. 10171–10185.
>
> [5] J. Wang et al. "A novel formulation of trace ratio linear discriminant analysis". In:
> IEEE Trans. Neural Netw. Learn. Syst. 33.10 (2021), pp. 5568–5578.
>
> [6] K. He et al, "Deep residual learning for image
> recognition," In: Proc. 29th IEEE Conf. Comput. Vis. Pattern Recognit., 2016, pp. 770–778.

---

> > ### Comment · Reviewer_8jLu · 2026-06-19
> >
> > Thank you for the revisions on computational costs and hyperparameter selection, and for the additional experiments. I believe that the additional results have strengthened the paper, and therefore most of my concerns have been addressed (although the remark that the gains in the classification setting are marginal still stands).

---

> > > ### Author Response · Authors · 2026-06-19
> > > **Response to Reviewer 8jLu**
> > >
> > > Thank you very much for your positive assessment. We are pleased that the additional experiments and revisions have addressed most of your concerns. Regarding the performance gains, building on our current experiments applying the proposed methods to deep features, we plan to further incorporate the neighborhood-purification idea into deep architectures, for example, by designing a differentiable subspace-learning module or an auxiliary loss, thereby further enhancing the effectiveness of our methods.
> > >
> > > Thank you again for your constructive feedback, which has helped us substantially improve the manuscript.

---

### Review · Reviewer_W3Wr · 2026-04-30

**Summary Of Contributions:**

This paper presents a new supervised dimensionality reduction method, NePuDA, which seeks a subspace where each sample's local neighborhood is "purified", i.e., composed predominantly of same-class points. The idea is intuitively appealing and addresses limitations of existing LDA variants that rely on class-level or average neighborhood statistics. The paper includes a well-structured theoretical analysis (convergence, complexity) and thorough experiments on synthetic and real-world datasets. The method consistently outperforms twelve baseline DR methods on most benchmarks.

However, several aspects of the methodology, evaluation, and presentation require further clarification and strengthening. My detailed comments follow.

**Audience:**

Yes

**Audience Explanation:**

At least some individuals in TMLR's audience would be interested in the findings of this paper. The paper proposes a novel supervised dimensionality reduction method called Neighborhood-Purifying Discriminant Analysis (NePuDA), which introduces the concept of “neighborhood purification”, finding a subspace where each data point is surrounded predominantly by same-class neighbors. This idea directly addresses known limitations of classical LDA and its variants, particularly in handling data with irregular class shapes, high intra-class variance, and class overlap. The work includes a thorough theoretical analysis (convergence proof and complexity analysis) and comprehensive experiments on 13 real-world and synthetic datasets, comparing against 12 baseline methods. These contributions are relevant to the TMLR community, which includes researchers working on linear and nonlinear dimensionality reduction, discriminant analysis, metric learning, and robust feature extraction. Therefore, the findings are likely to attract the attention of readers interested in foundational and practically effective DR techniques.

**Broader Impact Concerns:**

See Requested Changes.

**Claims And Evidence:**

Yes

**Claims Explanation:**

While the submission provides extensive empirical comparisons on thirteen real-world datasets and two synthetic cases, and the theoretical analysis includes convergence proofs and complexity discussions, several key claims are not yet fully substantiated by clear and convincing evidence.

First, the core claim that NePuDA solves a non-convex ratio objective via an SDP relaxation is presented as an effective optimization strategy, but the manuscript does not provide a rigorous gap analysis between the original non-convex problem (Eq. 8) and the relaxed SDP formulation (Eq. 15). It is unclear whether the solution obtained from the SDP solver always corresponds to a feasible projection matrix W and whether it achieves global optimality with respect to the original objective. Without this analysis, the claimed optimality of the learned subspace remains uncertain.

Second, the scalability claims are not backed by concrete runtime or memory measurements on the reported datasets. The complexity analysis is theoretical; actual wall-clock times and memory footprints under the experimental settings would provide much stronger evidence for the method’s practical applicability, especially given the known limitations of interior-point SDP solvers.

Third, the superiority of the neighborhood-purification concept over existing local-structure methods, such as LMNN, NCA, or recent contrastive-learning-based DR techniques, is not directly validated. The baselines, while representative of LDA variants, do not include these modern competitors, and thus the evidence for the claimed advancement over general local DR methods is insufficient.

Finally, certain notation (e.g., Eq. 2) and implementation details (e.g., standardization, CVX parameters, initialization of Z^(0) are either missing or ambiguous, which hinders the reproducibility of the reported results. Clearer presentation and additional experimental details are needed to make the evidence more convincing.

**Requested Changes:**

1. In Eq. (1) and Eq. (2), the pure neighborhood sets $\mathcal{N}_i$ and $\mathcal{N}_i^+$ are defined as subsets of $k$ nearest neighbors. It is not explicitly stated whether the point $\mathbf{x}_i$ itself is excluded from its own neighborhood. In many local DR methods, self is excluded to avoid trivial solutions. If self is included, the within-class scatter $\mathbf{S}_i$ would always contain a zero term, which might weaken the discriminative power. The authors should specify whether $\mathbf{x}_i$ is excluded and discuss the implications.

2. The reformulation in Eq.~(9) replaces $\mathbf{W}$ with $\mathbf{Z} = \mathbf{W}\mathbf{W}^\top$. However, the original objective (8) is non-convex, and the SDP relaxation in (15) may solve a convex approximation of the transformed problem. Does the solution of (15) always correspond to a feasible $\mathbf{W}$? More importantly, is the obtained $\mathbf{W}$ globally optimal for the relaxed problem, or only a stationary point? The authors should discuss the gap between the SDP relaxation and the original non-convex problem, and whether the final projection matrix $\mathbf{W}$ is guaranteed to be a global maximizer of (8).

3. The SDP-based solver has complexity $\mathcal{O}(\tau m_0^2 n_0^2)$ with $m_0 = D(D+1)/2 + 2$. For high-dimensional data (e.g., images with thousands of pixels), this becomes prohibitive. While the limitations section acknowledges this, the experiments are conducted on datasets where the original dimensionality is at most a few hundred (after PCA pre‑processing to retain $\ge 90\%$ variance). The authors should report the actual runtime and memory consumption on these datasets and explicitly state the practical upper limit on $D$ that the current implementation can handle. A discussion of possible approximations (e.g., using random projections or incremental SDP solvers) would also be beneficial.

4. The parameter sensitivity analysis (Fig.~6) shows that performance is stable across a range of $k$, but the tested values are relatively narrow (2--16). For datasets with larger class sizes, $k$ may need to be larger to purify neighborhoods effectively. Could the authors provide guidance on how to choose $k$ adaptively based on data characteristics, or discuss whether a heuristic (e.g., proportional to the minimum class size) is sufficient? Additionally, the two variants NePuDA and NePuDA$^+$ use different strategies for defining pure neighborhoods; a direct comparison of their behaviors under identical $k$ values would help understand their relative merits beyond the aggregated accuracy tables.

5. The primary evaluation metric is the classification accuracy of a 3‑NN classifier in the reduced subspace. While this is standard, it would be informative to report other measures that quantify the quality of the embedding itself, such as the normalized mutual information (NMI) or the silhouette score. Furthermore, the train/test split is 70/30 stratified by class, but the authors do not specify whether this split is performed in a stratified manner per class to maintain class proportions. Clarifying this and reporting variance across multiple random splits (already done with 5 trials) is good, but adding a statistical significance test (e.g., paired $t$-test) between the best baseline and NePuDA would strengthen the conclusions.

6. The baselines are well-chosen from classical DR, but the paper would benefit from comparisons with more recent local structure‑preserving methods, such as Large Margin Nearest Neighbor (LMNN), Neighborhood Components Analysis (NCA), or contrastive learning‑based DR (e.g., SupCon). While the focus is on linear DR, comparing with nonlinear manifold methods (e.g., $t$-SNE, UMAP) used in a supervised manner could illustrate the added value of the proposed linear method. If such comparisons are not feasible, the authors should explicitly define the scope and justify the chosen baselines.

7. he definition of $\mathcal{N}_i^+$ in Eq. (2) contains the notation $\{\mathbf{x}_i\}_{i=1,n}^{f=\ell_i}$,
which is non‑standard and potentially confusing. It is meant to indicate that the $k$ nearest neighbors are sought only among samples with label $\ell_i$. The authors should rewrite this clearly, e.g., ``$\mathbb{N}_k(\{\mathbf{x}_j \mid \ell_j = \ell_i\}, \mathbf{x}_i)$''. The current form may hinder reproducibility.

8.  In Fig. 5, the reduced subspace is further projected to 2D via PCA for visualization. This additional transformation can distort the actual separation achieved by the method. The authors should justify why direct visualization of the top two dimensions of the learned subspace is not used, or discuss how the PCA step may affect the interpretation. Ideally, the raw projection onto the first two dimensions of $\mathbf{W}$ should be shown, even if separation is not perfect.

9. The related work section should more thoroughly discuss connections between the proposed neighborhood-purification concept and recent advances in representation learning. Specifically, the following connections and references deserve attention: 1) The proposed method's emphasis on local discriminative structure shares conceptual similarities with contrastive learning frameworks and deep metric learning approaches. Recent work such as "Beyond Seen Bounds: Class-Centric Polarization for Single-Domain Generalized Deep Metric Learning" proposes polarization of class centers for robust metric learning, which is relevant to NePuDA's goal of ensuring local class purity in the subspace. 2) The robustness of NePuDA to noisy or overlapping class boundaries connects with research on learning with imperfect supervision. "From Calibration to Refinement: Seeking Certainty via Probabilistic Evidence Propagation for Noisy-Label Person Re-Identification" addresses uncertainty in learned representations, which parallels NePuDA's treatment of "at-risk" data points with mixed-class neighborhoods. 3) For potential extensions of NePuDA to multimodal or video-based applications, connections to recent work on spatio-temporal representation learning and multimodal fusion should be discussed. Relevant papers include "Diff-LMM: Diffusion Teacher-Guided Spatio-Temporal Perception for Video Large Multimodal Models" and "AIGC Video Detection Based on the Fusion of Spatial-Frequency-Optical Flow Multimodal Features", both of which leverage multimodal feature integration for robust discrimination. 4) The directional and structure-aware aspects of the proposed method may benefit from insights in "DAWN: Direction-aware Attention Wavelet Network for Image Deraining", which demonstrates the importance of directional awareness in feature extraction.
The authors should incorporate these discussions into the related work section and, where feasible, compare against methods inspired by these paradigms.

10.  Several implementation details are not provided:

1) The specific CVX solver parameters (e.g., precision, maximum iterations).
2) How the initial $\mathbf{Z}^{(0)}$ is generated (randomized? any specific distribution?).
3) Whether the data are standardized before applying DR.

    These details are important for reproducibility and should be included.

 Limited discussion on the failure cases or scenarios where NePuDA may underperform.
    The paper shows strong overall results, but there are a few datasets (e.g., Hayes‑roth, Balance‑scale) where NePuDA does not outperform the best baseline. A brief analysis of why the neighborhood‑purification strategy may struggle in these cases (e.g., data distribution, class overlap characteristics) would add valuable insight and help set realistic expectations for practitioners.

11. While generally clear, the manuscript contains minor grammatical errors and awkward phrasing (e.g., the projection results of the proposed method, NePuDA, with 6 baseline methods, the within‑class‑neighborhood scatter of $\mathbf{x}_i$ is defined as follows: $\mathbf{S}_i = \sum_{\mathbf{x}_j\in \mathcal{N}_i\mathrm{or}\mathcal{N}_i^+}(\mathbf{x}_i - \mathbf{x}_j)(\mathbf{x}_i - \mathbf{x}_j)^T$ lacks proper spacing). A thorough proofreading is recommended to improve readability.

---

> ### Author Response · Authors · 2026-06-16
> **Response to Reviewer W3Wr (1)**
>
> We are sincerely thankful for the reviewer’s careful evaluation of our manuscript. We also greatly appreciate the constructive feedbacks and valuable suggestions, expertly  provided on our methodology design, mathematical optimization, complexity analysis, and experimental designs. Below, we respond to the reviewer’s valuable feedback. All revisions made in the manuscript are highlighted in blue for easy identification.
>
> ---
>
> * Regarding pure neighborhood sets construction, we sincerely appreciate the reviewer's comment. In our design and implementation, for the pure neighborhood construction of $\mathbf{x}_i$, we exclude $\mathbf{x}_i$ itself, to avoid triviality and ensure sufficient candidates to construct the pure neighborhood. To make this clearer, we add descriptions in our manuscript.
>
>   **Revision:** We clarify that we exclude $\mathbf{x}_i$ itself when constructing its pure neighborhood set, in highlighted part of Section 2.2 (Pure Neighborhood Construction), at the beginning of line 172.
> * Regarding the gap between the relaxed problem and the original objective, we sincerely appreciate the suggestions of the reviewer. Here we explain both the optimization objective and optimization constraints, respectively. First, for the optimization objective, the original optimization objective in Eq. (8) is in the form of fractional programming (FP). To solve the problem, we adopt the classical Dinkelbach's transform [1,2], under which the original fractional problem Eq. (12) can be equivalently solved through a sequence of parameterized subtractive subproblems in Eq. (13). Second, for the relaxation of the constraints, Eq. (15) is solved over the relaxed set $\Psi$ which is strictly larger than the original feasible set $\Phi$, i.e., $\Phi\subseteq \Psi$. After obtaining $\mathbf{Z}\in\Psi$, we recover the projection matrix $\mathbf{W}$ via eigen-decomposition by taking the eigenvectors associated with the largest $d$ eigenvalues. This recovered $\mathbf{W}$ always satisfies the orthogonality constraint of the original problem, and is therefore a feasible solution of the original problem. Since the relaxed subproblem at each iteration, given in Eq. (14), is convex, the SDP procedure in Eq. (15) yields a globally optimal solution to each subproblem. Consequently, the resulting iterative algorithm converges to a globally optimal solution of the relaxed problem~(12), rather than merely to a stationary point [10], and the solved $\mathbf{W}$ is recovered from global optimizer.
>
>   More importantly, to be precise, the gap between the global optimality of Eq. (8) and the solution obtained from Eq. (12) lies in whether the optimizer of Eq. (15) falls in $\Phi$ or not, which is essentially induced by the difference between the original feasible set $\Phi$ and its relaxed convex feasible set $\Psi$. If the optimizer falls in $\Phi$, we could obtain the globally optimal solution of the original problem. Besides, to minimize the likelihood that the relaxed optimizer falls outside $\Phi$, we adopt the smallest convex set containing $\Phi$ (i.e. the convex hull of $\Phi$) which is denoted by $\Psi$, to make the relaxation which induces optimality gap as tight as possible, thereby approximating the original constraints to the greatest extent.
>
>   **Revision:** We clarify that $\mathbf{W}$ is feasible for the original problem and globally optimal for the relaxed problem. We discuss the potential gap between the original problem and relaxed problem in highlighted part of Section 3 (Optimization), in terms of both objectives (lines 241-244) and constraints (lines 234-236).

---

> ### Author Response · Authors · 2026-06-16
> **Response to Reviewer W3Wr (2)**
>
> * Regarding the practical applicability and computational complexity of the proposed methods, we sincerely thank the reviewer for suggesting us reporting the empirical computational cost. To do so, we conduct an empirical study on the YaleB dataset using the two versions of our methods as a demonstration. Specifically, we investigate the upper bound of runtime with respect to both the number of training samples and the input dimensionality. To this end, we consider two sample sizes, i.e., 1,000 and 2,000. For each setting, the input dimensionality is traversed in \{50, 75, 100, 125, 150, 200\}, which is implemented by preserving certain number of principal components. We record the running time of both algorithms that they finish one trial. All experiments are conducted in MATLAB R2022b on a Linux server (Oracle Linux Server 8.10) equipped with 256 GB RAM. The runtime results of both methods are reported Table 5 in the manuscript. As observed, when the input dimensionality reaches the scale of several hundreds (which is still far below the available memory capacity), the runtime of both methods increases significantly and approaches nearly 20 hours or more, which can be regarded as the practical upper limit of the algorithms. Also, to better illustrate the difference between runtime of two methods, we further discuss the difference in time complexity of within-class-neighborhood scatter construction for both versions of proposed methods in Section 4.2. For NePuDA, the construction of $S_i$ is $\mathcal{O}\left(nk\right)$, whereas that of NePuDA$^+$ is $\mathcal{O}\left(\sum_{i=1}^C {n^{(i)}}^2\log n^{(i)}\right)$, which is more time-consuming than NePuDA.
>
>
>   To accelerate the optimization strategy and improve its capability to handle data with large sample size and high dimension, we aim to achieve some possible approximations in our future work. One possible direction is to optimize each column of $\mathbf{W}$ individually at each iteration, such that the optimization variable reduces to a vector at each step, rather than optimizing a matrix, thereby enhancing the computational efficiency and the ability in handling large-scale high-dimensional data. Based on these incremental strategies, some well-established optimization frameworks (e.g., constrained concave-convex procedure) for solving DC (difference-of-convex) programs can be adopted to obtain the solution. Despite favorable convergence properties and computational efficiency of these approximation methods, they may converge to a stationary point rather than the global optimum, which may lead to sub-optimal solutions. Therefore, these potential incremental optimization strategies are more suitable for scenarios involving massive training samples and extremely high-dimensional data. By contrast, for small-sample settings with higher accuracy requirements, the current optimization strategy is more preferable.
>
>   **Revisions:** We report the running time of both methods in Appendix K in the manuscript, and discuss the upper limit of the current implementations, with further discussions about the incremental approximations. We also further discuss the time complexity of both methods in Section 4.2, from line 294 to line 295.

---

> ### Author Response · Authors · 2026-06-16
> **Response to Reviewer W3Wr (3)**
>
> * Regarding the parameter sensitivity analysis, we are truly thankful for reviewer's insightful comment about the search range of large-scale datasets. We agree with the reviewer that for NePuDA, the search range should be broader for large-scale datasets. We first discuss the heuristic to determine the value of $k$, which is from the design of pure neighborhood construction. Specifically, we adopt the proportion to the minimum class size for both methods, and NePuDA requires a larger proportion compared to NePuDA$^+$, as we present in the second paragraph of Section 5.3.2 (Experimental Setup), and the exact value is determined by the 5-fold cross validation in the experiments. This is because NePuDA finds neighboring data points first and then picks data points with same labels, a relatively larger $k$ is required to ensure sufficient within-class neighbors are considered. For NePuDA$^+$, it starts from same-class data points and then selects neighboring data points, so a smaller $k$ is generally sufficient to build up the within-class-neighborhood scatter. The observations of synthetic results from Figure 4 in the manuscript further support this heuristic, as NePuDA could achieve the neighborhood purification at $k=80$, while NePuDA$^+$ requires less with $k=60$.
>
>   Besides, to enlarge the searching interval of large datasets and compare the behaviors of both methods under identical values of $k$, we broaden the search interval of NePuDA on YaleB dataset which has relatively large number of data points, as NePuDA requires larger $k$ in general. Specifically, we broaden its search range as \{6,12,18,24,30,36,42,48\} to cover both well-performing and less favorable parameter settings, following the same protocol adopted for both methods on the other datasets. The results on YaleB presented in Figure 6(c) and Figure 6(f) clearly illustrate the behaviors of both methods under identical values of $k$. NePuDA$^+$ outperforms NePuDA when $k$ is small (e.g., for $k=6$, the mean accuracy of NePuDA across reduced dimensions is 75.60\%, while for NePuDA$^+$, it's 85.39\%), whereas NePuDA becomes comparable to NePuDA$^+$ as $k$ increases, which supports the heuristics to determine $k$.
>
>   **Revisions:** We alter the original traverse range of NePuDA on YaleB dataset with larger range in sensitivity analysis part, and make comparisons of their behaviors under identical $k$ values to present their relative merits in highlighted part of Appendix F, from line 841 to line 845.
>
> ---
>
> * Regarding the evaluation metrics, we thank the reviewers for suggesting additional measures such as NMI and silhouette score. NMI measures the agreement between clustering assignments and ground-truth labels, reflecting global class-level consistency, while the silhouette score evaluates the geometric compactness and separation of embeddings. These clustering-based metrics effectively reflect inter-class separability, while may not directly capture neighborhood purity, which is the primary focus of our method. Our approach aims to improve local neighborhood purification to enhance discriminativeness of the subspace, while the class-level separation is not our target. Therefore, we find that these metrics may not be the best options to reflect the neighborhood purification effect.
>
>   To take the reviewer's valuable advice into consideration, we have made the following revisions to further strengthen our conclusions. Specifically,
>
>   * We clarify that the train/test split is stratified by class to maintain class proportions.
>   * We conduct a statistical significance test to demonstrate the effectiveness of NePuDA$^+$. Specifically, we first select the strongest baseline by averaging the classification accuracy of all baseline methods across the 13 real-world datasets in Tables 1 and 2, and then conduct paired $t$-tests with NePuDA$^+$ in terms of mean accuracy across datasets. The results show that ANMMP achieved the highest average accuracy among all baselines, and NePuDA$^+$ significantly outperforms ANMMP (mean improvement = 1.95\%, p-value = 0.0043) at the 1\% significance level.
>
>   **Revisions:** We clarify that the train/test split is made in a stratified manner to maintain class proportions in Section 5.3.2 (Experimental Setup) at line 432, and the statistical test is provided in Section 5.3.3 (Classification Performance Comparison), from line 463 to line 466.

---

> ### Author Response · Authors · 2026-06-16
> **Response to Reviewer W3Wr (4)**
>
> * Regarding the baselines, we sincerely appreciate the reviewer's recommendations to make the comparison more comprehensive. Here we include LMNN [3] and NCA [4] as the new baselines for comparison. Specifically, for NCA, we implement it by directly optimizing a linear projection matrix using a quasi-Newton method for fast convergence. For large datasets (e.g., YaleB), we alternatively adopt a conjugate gradient based optimizer to reduce the memory overhead, despite of a slower convergence speed. The projection is initialized randomly and trained for a fixed number of iterations with an $\ell_2$ regularization term to avoid overfitting. For LMNN, we adopt a $k$-nearest neighbor scheme to define target neighbors (with $k=3$) and optimize the projection matrix using a gradient-based approach. We observe that NePuDA$^+$ generally outperforms both methods, mainly because its min-max strategy purifies the neighborhoods of all data points by emphasizing the worst-case samples rather than treating all data points uniformly.
>
>   For the scope of the baselines, since our DR methods are developed in a linear setting, the scope of selected baselines is within linear DR methods, to ensure a fair comparison. We agree with the reviewer that the comparison with nonlinear DR methods will make the evaluation more comprehensive and convincing. Therefore, in future work, we plan to extend our methods to nonlinear settings and compare them with these approaches, as we discuss by the end of Section 6 (Conclusions).
>
>   **Revisions:** We add NCA and LMNN in the baselines and present their experimental results in Table 1 and Table 2 of Section 5.3.3 (Classification Performance Comparison). We also clarify the scope of the baselines is linear DR methods in Section 5.1 (Baseline Methods), at line 308.
>
> ---
>
> * Regarding the notations in Eq. (2), we thank the reviewer for the insightful suggestion. For pure neighborhood $\mathcal{N}^+_i$ of NePuDA$^+$, different from $\mathcal{N}_i$ of NePuDA, our intention is to select nearest neighbors of $\mathbf{x}_i$ from data points sharing the same label with it. Therefore, the current notation exactly means that $\mathcal{N}_i^+$ is constructed by $\mathbf{x}_i$'s $k$ nearest neighbor which sought among samples with label $\ell_i$, and our intended meaning is consistent with the reviewer’s interpretation.
>
> ---
>
> * Regarding the visualization in Figure 5, we follow the reviewer's suggestion and visualize the projection results using the top two dimensions of the learned subspace for each method. The visualization is provided at [https://anonymous.4open.science/r/Further-information-of-NePuDA-0D4F/](https://anonymous.4open.science/r/Further-information-of-NePuDA-0D4F/). We observe that the results obtained using the first two dimensions exhibit a trend similar to those produced by PCA, although our method shows slightly greater class overlap in the former case. One possible reason is that although the first two dimensions capture the majority of the discriminative information, they may not contain all of it. The remaining dimensions of the learned subspace can still carry useful discriminative signals. In contrast, applying PCA to this learned subspace might be able to aggregate information across all dimensions, thereby producing a similar yet slightly better visualization for class discrimination.
>
> ---
>
> * Regarding the related work, we sincerely appreciate the reviewer for introducing several existing works that share conceptual connections with our proposed methods. We strongly agree with the reviewer that these recent advances in representation learning are closely related to the motivation and intuition behind the proposed neighborhood purification strategy. Therefore, we discuss them further in Appendix C.5, as Appendix C is primarily for connecting our methods to existing methods. In recent studies, Centerpolar [5] serves as a representative contrastive learning and deep metric learning method which aims to enhance local discriminability by polarization of class centers, which shares intuition with the neighborhood purification mechanism in NePuDA. Recent studies on noisy-label learning and uncertainty-aware representation refinement like CARE [6], further highlight the importance of handling ambiguous samples, which is also one of the main motivations of our proposed methods. In addition, neighborhood purification proposed in our work could also be extended to several promising directions for handling more complex scenarios, such as spatio-temporal perception [7], video detection [8], and direction-aware feature extraction [9].
>
>   **Revisions:** We discuss the relationship between recent representation learning and our work in Appendix C.5.

---

> ### Author Response · Authors · 2026-06-16
> **Response to Reviewer W3Wr (5)**
>
> * Regarding the implementation details, we thank the reviewer's reminders on clarifying them to improve the reproducibility. Here, we supplement the discussion with additional implementation details.
>   1. Specifically, we use CVX with the SDPT3 solver under the default CVX precision level and the default SDPT3 stopping settings, without manually tuning solver tolerances or the maximum number of iterations.
>   2. The initial $\mathbf{Z}^{(0)}$ is randomly initialized with entries drawn independently from a uniform distribution over $(0,1)$, serving as the starting point of the iterative optimization procedure.
>   3. The proposed methods do not require standardization as a necessary step to avoid the potential influence on the recognition of riskiest neighborhoods.
>
>   For scenarios that NePuDA does not achieve the best performance, we appreciate the reviewer's insightful observation. NePuDA constructs pure neighborhoods by first selecting geometric nearest neighbors and then retaining only those with the same label. It is therefore particularly effective on complex datasets with class overlap, provided that local regions contain sufficient same-class samples to form reliable pure neighborhoods. This is consistent with its strong performance on Yeast, Sonar, Iris, and JAFFE. However, when local neighborhoods are heavily mixed and contain only a few same-class samples, the resulting pure neighborhoods may become insufficient or unstable. To address this issue, we further introduce NePuDA$^+$, which constructs pure neighborhoods more reliably and achieves greater effectiveness and robustness across diverse complex scenarios.
>
>   **Revisions:** We clarify implementation details in highlighted part of Section 5.3.2 (Experimental Setup) from line 421 to line 424. Meanwhile, we further discuss the limitations of NePuDA in Section 5.3.3 (Classification Performance Comparison), which are highlighted in the manuscript from line 467 to line 475.
>
> ---
>
> * Regarding the grammar and phrasing issues, we sincerely thank the reviewer for the effort in thorough reading. For the kindly indicated examples, we rewrite the expression as \textit{We compare the proposed methods (NePuDA and NePuDA$^+$) with 6 baseline methods from all three main categories}, and we add proper spacing for the formulae in the manuscript. The proofreading is made on the entire manuscript, including both the original and newly added content.
>
>   **Revisions:** We proofread the entire manuscript, including both the original and newly added content, and correct all the identified grammatical and phrasing issues.
>
>  ---
>
> **References:**
>
> [1] W. Dinkelbach. ``On nonlinear fractional programming". Manage. Sci., vol. 13, no. 7, pp. 492–498, Mar. 1967.
>
> [2] S. Schaible. ``Fractional programming. II, on Dinkelbach’s algorithm". Manage. Sci., 22(8): 868–873, 1976.
>
> [3] K. Q. Weinberger and L. K. Saul. ``Distance metric learning for large margin
> nearest neighbor classification." In: J. Mach. Learn. Res. 10.2 (2009).
>
> [4] J. Goldberger et al. ``Neighbourhood components analysis". In: NeurIPS 17 (2004).
>
> [5] X. Yuan et al. ``Beyond Seen Bounds: Class-Centric Polarization for Single-Domain Generalized Deep Metric Learning". In: arXiv preprint arXiv:2601.09121 (2026).
>
> [6] X. Yuan et al. ``From Calibration to Refinement: Seeking Certainty via Probabilistic Evidence Propagation for Noisy-Label Person Re-Identification". In: arXiv preprint arXiv:2602.23133 (2026).
>
> [7] J. Dang et al. ``Diff-LMM: Diffusion Teacher-Guided Spatio-Temporal Perception for Video Large Multimodal Models". In: Proc. 34th IJCAI. 2025, pp. 873–881.
>
> [8] H. Sheng et al. ``AIGC video detection based on the fusion of spatial-frequency-optical flow multimodal features". In: J. Syst. Eng. Electron. (2026).
>
> [9] K. Jiang et al. ``DAWN: Direction-aware attention wavelet network for image deraining". In: Proc. 31st ACM MM. 2023, pp. 7065–7074.
>
> [10] K. Shen, W. Yu. ``Fractional programming for communication systems—Part I: Power control and beamforming". In: IEEE Trans. Signal Process., 2018, 66(10): 2616-2630.

---

### Review · Reviewer_6H8h · 2026-06-02

**Summary Of Contributions:**

This work studies supervised dimension reduction via Linear Discriminant Analysis (LDA). The authors argue that previous LDA methods suffer from at-risk data points at mixed class neighborhoods, and propose Neighborhood-Purifying Discriminant Analysis (NePuDA) that to find a suitable subspace, such that each data point is enveloped by neighbors belonging to the same class. They also conduct experiments to demonstrate the usefulness of the proposed method.

**Audience:**

No

**Audience Explanation:**

Most of the applications and tasks studied in this work are relatively outdated, and mostly restrained to low-dimensional. It's unclear what realistic applications can the proposed work be applied to.

**Claims And Evidence:**

No

**Claims Explanation:**

1. The first contribution may not hold. As discussed in the existing works, the concept resembles multiple existing works such as NMMP, LFDA, etc.. It requires specific discussion regarding the differences from those works when deriving the objective of NePuDA.

2. The third contribution may not hold. Among the datasets, only the face-recognition datasets are relatively high-dimensional, which has already been processed to a lower dimension by PCA.

3. NePuDA actually does not outperform lots of baselines in experiments. More discussion is required.

4. The best performing variant, NePuDA+, seems to conflict with the motivation, which still imposes stronger intra-class distances, as previous methods.

5. Significance analysis is lacking, as most of the improvements are within the standard deviations.

6. Some claims seem too strong:
- "More recently, deep models, including deep neural networks and large language models, have demonstrated their strength in representation learning. Integrating DR with these models can further boost their performance for several key reasons.", but none of the follow-ups discuss the application to LLMs.

**Requested Changes:**

Please find the details in the sections above.

---

> ### Author Response · Authors · 2026-06-16
> **Response to Reviewer 6H8h (1)**
>
> We sincerely thank the reviewer for the thorough evaluation of our manuscript and greatly appreciate the insightful and constructive comments on the contributions, experimental discussions, and claims presented in our work. We provide detailed responses to each comment below. For clarity and ease of reference, all revisions made to the manuscript are highlighted in blue.
>
> ---
>
> * Regarding the first contribution, we appreciate the reviewer's insightful comments on differentiating our method from existing neighborhood-level approaches, particularly those based on the concept of pure neighborhoods. We agree that our original description did not clearly articulate this contribution. Indeed, several existing methods (e.g., LMNN and LFDA) already emphasize focusing on the same-label neighbors.
>
>   The key distinction lies in our integration of a min-max strategy with neighborhood purification. By explicitly targeting the most difficult-to-purify points, our approach improves the lower bound of neighborhood purity across all data points, thereby achieving more effective purification overall. In contrast, existing neighborhood exploration methods primarily rely on averaging-based criteria. For instance, LFDA minimizes the weighted sum of pointwise local within-class scatter matrices while maximizing the sum of between-class scatter matrices, and NMMP optimizes the ratio of such sums to minimize intra-class and maximize inter-class distances. When averaging is used as the learning objective, the resulting projections tend to be dominated by already well-separated or well-purified regions. This often preserves the original data arrangement in the input space and introduces only limited additional discriminative information.
>
>   By contrast, NePuDA and NePuDA$^+$ adopt a sample-wise worst-case perspective, focusing on the lower bound of neighborhood purity for all samples. As shown in Eq. (5) and the subsequent objective in Eq. (8), our formulation avoids arithmetic or weighted sums over all samples or local pairs. Instead, it prioritizes protecting the riskiest points by minimizing the largest trace of within-class neighborhood scatters. This min-max design explicitly purifies the most challenging neighborhoods, thereby enhancing the overall lower bound of neighborhood purity rather than being dominated by already well-purified regions. The superiority of our approach over existing neighborhood-level methods is demonstrated on synthetic datasets in Figures 1 and 4.
>
>   **Revisions:** To make the descriptions of contributions more precise, we have revised the first contribution of Section 1.1 (Our Contributions) in the manuscript accordingly.
>
> ---
>
> * Regarding the datasets, we are sincerely grateful for the reviewer's comments.
>
>   Regarding dataset dimensionality, the primary goal of our experiments is to evaluate whether the proposed methods can extract more discriminative features from the original data. Discriminative information may be obscured by redundancy and noise, even in relatively low-dimensional datasets. For example, although the Hayes-Roth dataset contains only five features, the $k$NN classifier achieves an accuracy of only $40.51\pm7.34$ on the raw data. In contrast, the proposed method NePuDA$^+$ significantly improves performance to $72.82\pm11.15$. This demonstrates that supervised DR remains meaningful and effective even for low-dimensional data.
>
>   We apply PCA as a pre-processing step on high-dimensional datasets to follow a common practice in the dimensionality reduction community [1,2,3], including several representative works published in TMLR [4,5]. This reduces computational burden while preserving most of the data variance and avoiding the loss of principal information. Importantly, PCA is applied consistently to all compared methods, ensuring a fair and unbiased comparison. Since PCA is unsupervised and does not utilize label information, the resulting representation may still exhibit significant class overlap and mixed local neighborhoods when the original data has a complex distribution. For example, on JAFFE dataset, if we directly use 3-NN as the classifier on the PCA-processed data, a poor performance 46.88$\pm$2.92 is obtained, which is significantly lower than the results with projections generated by supervised DR methods presented in our manuscripts (e.g., 88.54$\pm$1.80 for NePuDA$^+$). These results highlight NePuDA's superior capability in extracting highly discriminative features beyond what unsupervised pre-processing can achieve.

---

> ### Author Response · Authors · 2026-06-16
> **Response to Reviewer 6H8h (2)**
>
> * Regarding the performance of the first version, NePuDA, we thank the reviewer for the opportunity to let us clarify its applicability. NePuDA constructs the pure neighborhood by first selecting geometric nearest neighbors and then retaining those with the same label. Therefore, NePuDA is expected to be more effective for complex datasets with class overlap where local regions still contain sufficient same-class samples, allowing reliable pure neighborhoods to be constructed from the original geometric neighborhoods. In such cases, NePuDA can effectively purify local neighborhoods through the proposed min-max objective. Empirically, this is consistent with the results on datasets such as Yeast and Sonar (with best performance among all methods), Iris and JAFFE (with best or almost-best performance except NePuDA$^+$), where adequate same-class samples are provided in local neighborhoods. On the other hand, when the original neighborhood is severely mixed with points from different classes, and further contains relatively limited number of same-class samples, the pure-neighborhood structure constructed by NePuDA may become insufficient or less stable.
>
>   To address this issue, we further introduce NePuDA$^+$, which directly selects nearest neighbors from the same class to construct the pure neighborhood. This strategy provides a more stable and informative same-class local set for each sample. The superior classification performance of NePuDA$^+$ supports the effectiveness of the proposed neighborhood-purifying principle. Meanwhile, the performance difference between NePuDA and NePuDA$^+$ does not shift the focus away from the proposed objective. Instead, it indicates that pure-neighborhood construction defines the local structures to be purified, while the shared min-max objective for enforcing neighborhood purification across samples remains the key mechanism.
>
>   **Revisions:** We have further discussed the difference between NePuDA and NePuDA$^+$ in highlighted part of Section 5.3.3 (Classification Performance Comparison) of the manuscript from line 467 to line 475.
>
> ---
>
> * Regarding the possibly stronger intra-class distances caused by NePuDA$^+$, we are very grateful for the reviewers' observations and comments. Indeed, NePuDA$^+$ may result in larger intra-class distances. This is because our primary objective is not to enforce global intra-class compactness (i.e., minimizing overall intra-class distances). Instead, we focus on neighborhood purification, ensuring that each data point is locally surrounded by same-class neighbors in the learned subspace. Since our method does not explicitly constrain global intra-class distances, larger overall intra-class spreads can naturally occur. To illustrate this, we refer to the example in Figure 1 of the manuscript. As shown, our method successfully achieves neighborhood purification, which aligns with our core motivation. For the red class, our approach yields the largest intra-class distance (our method: 249.6414, WLDA: 197.3844, ANMMP: 100.2146, SDA: 227.8638). Nevertheless, it produces the most discriminative embedding among all compared methods. Thus, the potentially larger intra-class distance does not contradict our motivation; rather, it reflects our emphasis on local neighborhood purity over global compactness, leading to superior class separation.
>
> ---
>
> * Regarding the significance analysis, we sincerely thank the reviewer for the comments about providing a more promising evidence to recognize our method’s performance. To verify the statistical significance of NePuDA$^+$, we first identify the best-performing baseline by averaging classification accuracy across the 13 real-world datasets in Tables 1 and 2. ANMMP achieves the highest average accuracy among all baselines. Secondly, we conduct paired $t$-tests between ANMMP and NePuDA$^+$ in terms of mean accuracy across datasets, showing that NePuDA$^+$ significantly outperforms ANMMP with a mean improvement of 1.95\% and $p=0.0043<0.01$, indicating that the improvement of NePuDA$^+$ is statistically significant at the 1\% significance level.
>
>   **Revisions:** The result of the statistical test is provided in the highlighted part of Section 5.3.3 (Classification Performance Comparison) from line 463 to line 466.

---

> ### Author Response · Authors · 2026-06-16
> **Response to Reviewer 6H8h (3)**
>
> * Regarding the claim on relationship between DR and LLM research, we honestly thank the reviewer for reminding us to discuss DR applications to LLMs. Recent literature supports the claim that DR can effectively reduce complexity while improving both the effectiveness and computational efficiency of LLMs. For instance, DR techniques have been shown to enhance the compactness and efficiency of language model representations [6], compress high-dimensional LLM-generated embeddings for more effective and efficient downstream recommendation systems [7], and reduce activation dimensionality to enable model compression and faster inference [8]. In addition, DR can improve the quality of training data [9], which is critical for effective LLM training [10].
>
>   **Revisions:** We have expanded the discussion on how DR benefits LLMs in the highlighted part of the Introduction, from line 33 to line 41.-
>
> ---
>
> * The primary goal of this work is to demonstrate the superiority of our method in extracting highly discriminative features from complex data distributions. To ensure fair and reproducible evaluation, we followed standard experimental settings widely adopted in recent DR studies [2, 3, 4] as well as classical works [1, 11], including the use of PCA as a pre-processing step. To further validate the applicability of our approach in nonlinear settings and with deep neural networks, we conducted additional experiments on pre-trained deep features with various nonlinear downstream classifiers. The results are summarized in Appendix I of the manuscript, demonstrating the effectiveness of our methods beyond traditional linear scenarios and extending their potential for real-world applications. In future work, we plan to incorporate our proposed objective directly into deep architectures to enable end-to-end training and classification, further broadening the applicability of our methods.
>
>   **Revisions:** We have added experiments exploring the integration of our methods with deep neural networks in Appendix I. Additionally, in Section 6 (Conclusion), we have expanded the discussion on future directions, specifically incorporating our objective into deep architectures to form a unified end-to-end deep model (lines 530-533).
>
> **References:**
>
> [1] W. Bian and D. Tao. ``Max-min distance analysis by using sequential SDP relaxation for dimension reduction". In: IEEE Trans. Pattern Anal. Mach. Intell. 33.5 (2010), pp. 1037–1050.
>
> [2] J. Wang et al. ``Ratio Sum Versus Sum Ratio for Linear Discriminant Analysis". In: IEEE Trans. Pattern Anal. Mach. Intell. 44.12 (2022), pp. 10171–10185.
>
> [3] Z. Wang et al. ``Worst-Case Discriminative Feature Learning via Max-Min Ratio Analysis". In: IEEE Trans. Pattern Anal. Mach. Intell. 46.1 (2024), pp. 641–658.
>
> [4] M. M. Omati et al. ``A Max-Min Approach to the Worst-Case Class Separation Problem". In: Trans. Mach. Learn. Res. (2025).
>
> [5] H. V. Assel, C. Vincent-Cuaz, N. Courty, et al. ``Distributional Reduction: Unifying Dimensionality Reduction and Clustering with Gromov-Wasserstein". In: Trans. Mach. Learn. Res. (2025)
>
> [6] G. Zhang, Y. Zhou, and D. Bollegala. ``Evaluating unsupervised dimensionality reduction methods for pretrained sentence embeddings". In: Proc. LREC-COLING, pp. 6530–6543, 2024.
>
> [7] J. Ma, J. Qin, Y. Chen, Q. Feng, and S. Liu. Robust prototype-aware representation refinement for LLM-based sequential recommendation. Pattern Recognit., pp. 113550, 2026.
>
> [8] C. Sakr and B. Khailany. ``ESPACE: Dimensionality reduction of activations for
> model compression". In: NeurIPS 37 (2024), pp. 17489–17517.
>
> [9] F. Saberi-Movahed et al. ``Nonnegative matrix factorization in dimensionality reduction: A survey". In: ACM Computing Surveys 58.5 (2025), pp. 1–41.
>
> [10] C. Zhang et al. ``Harnessing diversity for important data selection in pretraining large language models". In: ICLR 2025, pp. 72980–73003.
>
> [11] K. Q. Weinberger and L. K. Saul. ``Distance metric learning for large margin
> nearest neighbor classification." In: J. Mach. Learn. Res. 10.2 (2009).

---

> > ### Comment · Reviewer_6H8h · 2026-06-19
> >
> > Thank the authors for the responses. However, several of my concerns remain hold, specifically about the motivation and key contributions of this work:
> > - Can the authors provide concrete evidence for the justification of NePuDA+ ? Since NePuDA+ is the only variant that can outperform the baselines.
> > - The significance analysis is performed wrt. weak baselines.
> > - The applicability to high-dimensional data remains limited. Especially, will the application of PCA break the motivation of NePuDA for high-dimensional data?

---

> > > ### Author Response · Authors · 2026-06-23
> > > **Response to Reviewer 6H8h (1)**
> > >
> > > * Regarding the first question, we appreciate the reviewer for the opportunity to further discuss the superiority of NePuDA$^+$. Although NePuDA and NePuDA$ ^+$ share the key idea of neighborhood purification through a min-max strategy, NePuDA$ ^+$ constructs a more stable and informative same-class local set for each sample by directly selecting its nearest neighbors from samples of the same class. This design extends NePuDA to more complex scenarios, particularly when the original geometric neighborhood is heavily mixed with samples from different classes and contains only a limited number of same-class neighbors. On top of the synthetic datasets for validating the effectiveness of both proposed methods in the manuscript, we further provide a simple illustrative example to further justify the motivation and effectiveness of NePuDA$ ^+$.
> > >
> > >   As shown in the example at https://anonymous.4open.science/r/Justifications-of-NePuDA_plus-D9BD/, the data exhibit a relatively complex distribution, with irregular class shapes and substantial overlap between Classes 2 and 3. We further observe that several samples from Class 3 are surrounded by samples from Class 2. NePuDA first identifies the geometric neighborhood of each sample and then retains only the same-class samples to construct its pure neighborhood. Under this setting, it may construct insufficiently informative pure neighborhoods for these Class 3 samples and consequently learn an almost horizontal projection direction, resulting in substantial class overlap. By contrast, NePuDA$^+$ first restricts the candidate set to same-class samples and then selects the nearest neighbors, thereby ensuring a more informative pure neighborhood and achieving more effective neighborhood purification.
> > >
> > >   In our experiments, NePuDA demonstrates the effectiveness of neighborhood purification on real-world datasets such as Yeast, Sonar, Iris, and JAFFE. For more complex data distributions, NePuDA$^+$ achieves more competitive performance by constructing more informative and stable pure neighborhoods. These results demonstrate the effectiveness of the neighborhood purification, while highlighting the importance of pure neighborhood construction for different data characteristics.
> > >
> > > ---
> > >
> > > * Regarding the second question, we thank the reviewer for the opportunity to further clarify our statistical analysis. Specifically, we select the baseline that achieves the highest average classification accuracy across all 13 datasets to conduct the statistical analysis. The mean accuracies of the baseline methods are as follows: **ANMMP: 78.90**, RSLDA: 78.39, $\ell_{2,1}$-LDA: 78.31, HMMDA: 78.07, WLDA:78.04, WDDR: 77.87, LMNN: 77.72, RLDA: 77.31, NCA: 76.81, FastSDA: 76.54, TRLDA: 74.56, LDA: 74.43, FSPCA: 67.07, PCA: 54.60. ANMMP therefore serves as the strongest baseline. Accordingly, we conduct the paired $t$-test between ANMMP and NePuDA$^+$ across the 13 datasets. The results demonstrate that NePuDA$^+$ significantly outperforms ANMMP, confirming its statistically significant superiority over the strongest baseline.

---

> > > ### Author Response · Authors · 2026-06-23
> > > **Response to Reviewer 6H8h (2)**
> > >
> > > * Regarding the third question, we sincerely thank the reviewer for more discussions on the applicability and the PCA processing.
> > >
> > >   To further evaluate the effectiveness of neighborhood purification on high-dimensional data, we replace the original SDP-based optimization framework with Riemannian gradient ascent applied to the objective function in Eq. (8) over the Stiefel manifold $St(D,d)$, defined by $ \left\lbrace \mathbf{W}\in\mathbb{R}^{D\times d} \mid \mathbf{W}^{\top}\mathbf{W}=\mathbf{I}_d \right\rbrace$, to improve the efficiency of our methods in directly processing the original high-dimensional data without PCA pre-processing. Specifically, we first derive the Euclidean gradient of the objective function in Eq. (8) and project it onto the tangent space of the Stiefel manifold to obtain the corresponding Riemannian gradient. A retraction operation is then applied after each update to ensure that the resulting solution remains feasible.
> > >
> > >   We evaluate the effectiveness of NePuDA$^+$ with the more efficient optimization strategy on high-dimensional data by comparing it with three representative baselines (TRLDA, HMMDA, and ANMMP), each selected from one of the three major categories of LDA variants. We conduct the experiment on the Yale dataset, whose original feature dimensionality is 1,024 (32 $\times$ 32), and the candidate subspace dimensionality is varied from 10 to $ 10\times \left \lfloor \frac{n_{train}}{10}  \right \rfloor $ with a step size of 10, where $n_{train}$ denotes the number of training data. To ensure a fair comparison, no PCA pre-processing is applied to any methods. The mean classification accuracy, its corresponding standard deviation, and the subspace dimensionality yielding the highest mean accuracy are reported for each method below. For the Riemannian optimization of NePuDA$^+$, to reduce the risk of convergence to an unfavorable local optimum, we adopt a multi-start initialization strategy and select the solution achieves the highest value of the objective in Eq. (8), evaluated by the training data. We adaptively determine the learning rate through Armijo backtracking line search, with an initial value of 1 and a contraction factor of 0.5. The iterative procedure is terminated when the Frobenius norm of the Riemannian gradient falls below $ 10^{-6}$, with a maximum iteration number is 500. All other experimental settings remain the same as those described in the manuscript.
> > >
> > >   | Methods    | AvgAcc $\pm$ Std | Opt. Dim. |
> > >   | :--------- | :--------------: | :-------: |
> > >   | TRLDA      |  $ 60.44 \pm3.30$  |   $d=50$  |
> > >   | HMMDA      |  $ 63.56 \pm8.69$  |  $d=110$  |
> > >   | ANMMP      |  $ 62.67 \pm3.65$  |  $d=100$  |
> > >   | NePuDA$^+$ |  $ 67.11 \pm2.86$  |  $d=110$  |
> > >
> > >   As shown by the results, NePuDA$ ^+$ continues to outperform the representative baselines, providing additional evidence of its effectiveness in extracting discriminative features directly from high-dimensional data and its potential applicability to such settings. This observation is consistent with that observed under PCA pre-processing. Therefore, we empirically validate that the PCA-processing may not undermine the motivation of neighborhood purification, for given input representation including high-dimensional data. In addition, we also observe that the methods with PCA pre-processing generally achieve higher classification accuracy than those without PCA pre-processing. This result suggests that PCA may help reduce noise and redundant information while preserving the principal structure of high-dimensional data, thereby facilitating better classification performance.
> > >
> > >   We also appreciate the reviewer’s valuable comment and agree that the applicability of the current method to raw high-dimensional data could be further improved. In future work, we plan to incorporate the neighborhood-purification idea into deep architectures, for example, by developing it as a differentiable module for discriminative feature learning. This would enable an end-to-end classification framework and further improve its applicability to high-dimensional data and real-world tasks.

---

> > > > ### Comment · Reviewer_6H8h · 2026-06-26
> > > >
> > > > Thank you for the explanation. However, my first question was about the conflicts between the proposed mechanism in NePuDA+ and the motivation of the paper. The toy example only shows the advantages of NePuDA+ over NePuDA. As NePuDA+ is the best-performed method, it's unclear whether the motivation of this work still holds.
> > > >
> > > > Regarding the second one, the significance of the improvements remains limited.
> > > >
> > > > Regarding the third one, the new results provide proper justification. Thank you.

---

> > > > > ### Author Response · Authors · 2026-06-27
> > > > > **Response to Reviewer 6H8h**
> > > > >
> > > > > We sincerely thank the reviewer for the follow-up questions and comments.
> > > > >
> > > > > ---
> > > > >
> > > > > * Regarding the first question, we sincerely appreciate the reviewer’s comment about further clarification of NePuDA$^+$.
> > > > >   The motivation of the proposed methods is to learn a subspace in which label information is used to establish class-homogeneous local structure for individual data points, particularly the most at-risk ones, through a min-max strategy, rather than optimizing an average criterion over all data points within each class. In this way, we aim to improve the lower bound of neighborhood purity in the learned subspace.
> > > > >
> > > > >   We acknowledge the observation that NePuDA$^+$ is the best-performed method. The reason is that, NePuDA$^+$ adopts a more effective pure neighborhood construction for more complex scenarios, based on the pre-restriction strategy which selects nearest neighbors from same-class points. This provides each data point with an adequately sized, robust, and informative pure neighborhood, thereby realizing neighborhood purification more effectively and aligning more closely with our original motivation in challenging settings exhibiting severe class overlaps with limited same-class neighbors in the original space, as illustrated in our toy example. Therefore, NePuDA$^+$ achieves generally superior performance based on the motivation of neighborhood purification and advanced pure neighborhood construction on complex data distributions.
> > > > >
> > > > > ---
> > > > >
> > > > > * We thank the reviewer for the second comment regarding the performance improvements. To provide a more comprehensive statistical evaluation, we conduct paired $t$-tests between NePuDA$^+$ and each baseline across the 13 datasets, to more rigorously assess whether the improvements achieved are statistically significant. The results are summarized below.
> > > > >   | Methods    |&nbsp;&nbsp;&nbsp;&nbsp;&nbsp;&nbsp;&nbsp;&nbsp;p-value   | Significance at 5% | Significance at 1% |
> > > > > | :--------- | :---------: | :-------: |:-------: |
> > > > > |PCA | $ 0.0007418$ | 1 |  1|
> > > > > |FSPCA | $ 0.0052068$ |1| 1|
> > > > > |LDA |  $ 1.2419078\times 10^{-5}$| 1 |  1|
> > > > > |FastSDA | $ 0.0023472$ | 1 |  1|
> > > > > |RSLDA   | $ 0.0009548$ | 1 |  1|
> > > > > |TRLDA   | $ 0.0084018$ | 1 |  1|
> > > > > |RLDA   | $ 0.0010223$ | 1 |  1|
> > > > > |$\ell_{2,1}$-LDA  | $ 0.0037534$ | 1 |  1|
> > > > > |WLDA| $ 0.0002969$ | 1 |  1|
> > > > > |HMMDA| $ 0.0034929$ | 1 |  1|
> > > > > |WDDR| $ 0.0180235$ | 1 |  0|
> > > > > |NCA| $ 3.9523476\times 10^{-5}$ | 1 |  1|
> > > > > |LMNN| $ 0.0008828$ | 1 |  1|
> > > > > |ANMMP| $ 0.0042997$ | 1 |  1|
> > > > >
> > > > >   As shown in the table, the performance improvements achieved by NePuDA$^+$ over all baselines are statistically significant at the 1\% significance level, except for WDDR, whose p-value slightly exceeds the 1\% threshold but remains significant at the 5\% level. Nevertheless, we agree with the reviewer that, despite the consistent improvements over all baselines, there remains room for further improvement. In future work, we plan to explore more advanced designs, such as integrating the neighborhood-purification mechanism into deep neural networks, to further enhance the performance of our method.